# Rapid metabolism fosters microbial survival in the deep, hot subseafloor biosphere

F. Beulig[1], F. Schubert [2], R. R. Adhikari [3], C. Glombitza [4], V. B. Heuer [3], K.-U. Hinrichs[3], K. L. Homola [5], F. Inagaki [6,7], B. B. Jørgensen[1], J. Kallmeyer [2], S. J. E. Krause[8], Y. Morono [6], J. Sauvage[5,10], A. J. Spivack[5] & T. Treude [8,9✉]

A fourth of the global seabed sediment volume is buried at depths where temperatures exceed 80 °C, a previously proposed thermal barrier for life in the subsurface. Here, we demonstrate, utilizing an extensive suite of radiotracer experiments, the prevalence of active methanogenic and sulfate-reducing populations in deeply buried marine sediment from the Nankai Trough subduction zone, heated to extreme temperature (up to ~120 °C). The small microbial community subsisted with high potential cell-specific rates of energy metabolism, which approach the rates of active surface sediments and laboratory cultures. Our discovery is in stark contrast to the extremely low metabolic rates otherwise observed in the deep subseafloor. As cells appear to invest most of their energy to repair thermal cell damage in the hot sediment, they are forced to balance delicately between subsistence near the upper temperature limit for life and a rich supply of substrates and energy from thermally driven reactions of the sedimentary organic matter.

[1] Center for Geomicrobiology, Department of Bioscience, Aarhus University, Ny Munkegade 114, 8000 Aarhus C, Denmark. [2] GFZ German Research Center for Geosciences, Section 3.7 Geomicrobiology, Telegrafenberg, 14473 Potsdam, Germany. [3] MARUM-Center for Marine Environmental Sciences, University of Bremen, Leobener Strasse 8, 28359 Bremen, Germany. [4] Department of Environmental Systems Science, ETH Zürich, Universitätstrasse 16, 8092 Zürich, Switzerland. [5] Graduate School of Oceanography, University of Rhode Island, Narragansett Bay Campus, 215 South Ferry Road, Narragansett, RI 02882, USA. [6] Kochi Institute for Core Sample Research, Japan Agency for Marine-Earth Science and Technology (JAMSTEC), Nankoku, Kochi 783-8502, Japan. [7] Mantle Drilling Promotion Office, Institute for Marine-Earth Exploration and Engineering, JAMSTEC, Yokosuka 237-0061, Japan. [8] Department of Earth, Planetary and Space Sciences, University of California Los Angeles, Los Angeles, CA 90095, USA. [9] Department of Atmospheric and Oceanic Sciences, University of California Los Angeles, Los Angeles, CA 90095, USA. [10] Present address: Department of Marine Sciences, University of Gothenburg, Carl Skottsbergs Gata 22B, 413 19 Göteborg, Sweden. ✉email: ttreude@g.ucla.edu

Organic matter buried in the seabed provides a continuous supply of carbon and energy for microorganisms, allowing their subsistence in sediments that are more than 100 million years old[1] and buried as deep as ~2.5 km below the seafloor[2]. It is argued that life in the subseafloor becomes progressively more challenging with depth due to a decrease in biodegradability of the remaining organic matter[3] and limitations associated with chemical diffusion rates and reaction affinities[4]. The energy supply for all life processes thereby drops by orders of magnitude. Furthermore, increasing temperature due to sediment burial, typically by about 30 °C per km, causes additional challenges for microbial life. Thermally-driven damage to vital biomolecules accelerates when temperature increases and forces microorganisms to increasingly invest their metabolic resources in maintaining structural and biochemical integrity[5]. For the deep and usually energy-depleted sedimentary biosphere, 80 °C has been proposed to be an upper thermal barrier for life[6–8]. This contrasts with high-energy hot environments, such as seafloor hydrothermal systems, which can support microbial populations up to 110 °C[9], and possibly even higher[10,11]. However, the recent detection of microbial cells combined with isotope systematics of biological metabolites in up to 120 °C hot sediments of the Nankai Trough subduction zone[12] indicates that the habitability of the marine subsurface extends far beyond the proposed 80 °C thresholds.

The effects of temperature on biotic and abiotic processes in marine sediment are complex and incubation experiments have shown that these processes change over narrow temperature intervals[13]. Below 50 °C, biogeochemical processes are primarily influenced by microbial degradation of organic matter. Acetate is a central metabolic intermediate in the anaerobic microbial food chain and accounts for nearly half of all carbon that is mineralized by the dominant terminal degradation processes, i.e., sulfate reduction and methanogenesis[14–16]. Above 50 °C, sediment geochemistry is increasingly subjected to and eventually dominated by the abiotic, thermal transformation of organic matter, thereby overprinting signs of biological activity. Thermochemical transformation of recalcitrant organic matter, otherwise resistant to microbial consumption, can generate acetate, petroleum, and volatiles such as methane and $H_2$[7,8,17]. These utilizable substrates, in principle, could sustain a deep, hot subseafloor biosphere over geologic timescales[7,8].

To investigate the upper-temperature limit of microbial life in the deep subseafloor, IODP (International Ocean Discovery Program) Expedition 370 drilled and recovered sediment cores in the Nankai Trough subduction zone off Cape Muroto, Japan (Site C0023; 32°22.00′N, 134°57.98′E; 4776 m water depth), using the drilling vessel (DV) Chikyu[18]. The steep, geothermal gradient at this site resulted in a temperature of 120 ± 3 °C at 1177 m below seafloor (mbsf), the deepest core retrieved from the basement[12,18] (Fig. 1). Despite the high temperatures, microbial cells were detected almost throughout the entire sediment column, albeit at extremely low concentrations of <500 cells per $cm^3$ in sediment above ~50 °C[12]. The remarkable biogeochemistry of Site C0023 featured a sulfate-poor but methane-rich zone above 600 mbsf, followed by a ~200 m transition into a deep sulfate-rich, methane-poor zone that persisted below 800 mbsf and was influenced by diffusive exchange with the underlying oceanic basement rock[18,19]. In the methane-rich zone, high molar ratios of methane over ethane ($C_1/C_2 > 10^3$) and $\delta^{13}C$ values of methane (–60‰ to –65‰) suggest that methane is microbially produced[12,18,20]. In the methane-poor, sulfate-rich zone $\delta^{13}C$-values of methane increase, and $C_1/C_2$ ratios steadily decrease with depth, indicative of a thermogenic source[12]. The thermal history of Site C0023 is complex, and involved heating associated with rapid burial over the past 400,000 years and

temperature increasing at depth from 60 to 120 °C, as well as short, localized heating events (up to 200 °C within fracture zones for less than 1 year) due to hot fluid incursions[19]. Despite the low organic carbon content ($C_{org} < 0.5$ wt%; Fig. 1b), acetate concentrations increased to exceptionally high levels (up to 11.6 mmol $L^{-1}$) below 600 mbsf, possibly due to thermal transformation of the organic matter at temperatures >60 °C. Deeper than 1030 mbsf, a drop in acetate to 3 mmol $L^{-1}$ and a concurrent increase in $\delta^{13}C$-acetate indicates microbial consumption toward the sediment-basement interface[12]. Combined, these observations were argued to confirm the existence of a deep subseafloor biosphere extending up to 120 °C[7,12,21]. However, it is not possible to unequivocally infer active metabolism of the observed cells based on observations of in-situ geochemical gradients as these are potentially relics of metabolism that occurred at lower temperatures in the past and have not yet been erased by diffusion.

In this work, we build on these initial findings and directly quantify the key energy-producing metabolic pathways in this environment through the application of highly sensitive radiotracer methods. We demonstrate that a small community of microorganisms with rapid metabolism subsisted in the deep and hot, subseafloor biosphere.

## Results and discussion

**Potential microbial sulfate reduction and methanogenesis in the deep, hot subseafloor biosphere**. We performed highly sensitive radiotracer experiments, near the theoretical limit of detection, to measure potential rates of sulfate reduction (using $^{35}S$-$SO_4^{2-}$) and methanogenesis (using $^{14}C$-dissolved inorganic carbon; DIC), and potential activity of anaerobic oxidation of methane (AOM, using $^{14}C$-$CH_4$) in samples recovered from deep in the sediment column (for methodical details of all procedures see Methods and Supplementary Material). All incubations were conducted close to in-situ temperatures (40, 60, 75, or 80 °C, and 95 °C; Fig. 1c). In total, well over 700 samples and controls were processed. Sediments were diluted with sterile, anoxic artificial seawater media to produce sediment slurries, and different electron donors were supplemented (trace $H_2$, acetate or methane), which may stimulate metabolic activity[22]. Reported microbial rates, thus, represent the potential metabolic activity of the extant microbial community in the sediment. Separate types of control incubations were performed with either live sediment, killed sediment, medium, or drill fluid to account for any contamination and/or non-biological radiotracer reaction (Fig. S1, Table S1). In AOM incubations, a high background of abiotic $^{14}C$-$CH_4$ conversion to $^{14}C$-$CO_2$ in the medium controls precluded detection of AOM activity in the 60 and 75 °C incubation, while some potential biological turnover was detected at 95 °C (Fig. S2).

Our findings demonstrate the prevalence of active methanogenic and sulfate-reducing populations in millions of years old sediment heated to extreme temperatures (up to ~120 °C). In our discrete samples, which span the range of sediment temperatures, microbial activity, based on $^{35}S$-$SO_4^{2-}$ and $^{14}C$-DIC radiotracer experiments, was consistently well above the background level of the control samples (0.09 pmol $CH_4$ $cm^{-3}$ $d^{-1}$ and 0.13 pmol $SO_4^{2-}$ $cm^{-3}$ $d^{-1}$; Fig. 1e–g). These activities were highest in the upper 320 mbsf with rates of ~$10^2$ to $10^3$ pmol ($CH_4$ or $SO_4^{2-}$) $cm^{-3}$ $d^{-1}$ in 40 °C incubations, similar to rates typical of shallow marine sediment[15,23]. In a relatively narrow sediment interval between 320 mbsf and 360 mbsf, where temperatures increase from 45 to 50 °C, sulfate reduction and methanogenesis rates dropped over three orders of magnitude. This drop coincides with a similarly steep drop in microbial cell

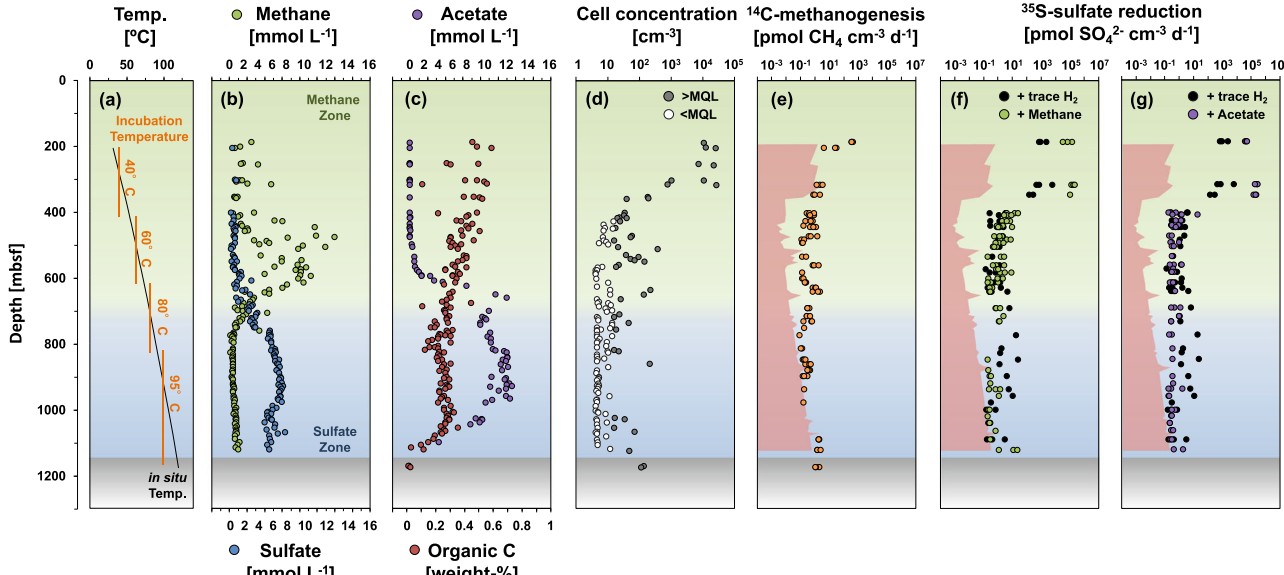

**Fig. 1 Temperature, biogeochemistry, cell counts, and anaerobic microbial processes in IODP Site C0023 sediment. a** Temperature of sediment and incubation experiments, **b** dissolved methane and sulfate, **c** dissolved acetate and bulk sediment organic carbon, **d** concentration of microbial cells fluorescently stained with SYBR green, and **e**–**g** potential rates of methanogenesis from DIC (0.68 mmol $L^{-1}$ in medium) with trace $H_2$ (130 nmol $L^{-1}$) added and sulfate reduction from sulfate (5 mmol $L^{-1}$ in medium) with either trace $H_2$ (130 nmol $L^{-1}$), acetate (5 mmol $L^{-1}$), or methane (100 vol% $CH_4$ in the headspace) added. Note that sulfate reduction + trace $H_2$ is presented both in panel (**f**) and (**g**) for better comparison. For more details on the experimental setup, please refer to the methods. The minimum quantification limit (MQL) for cell counts was 16 cells $cm^{-3}$. Data for methanogenesis and sulfate reduction is only shown for rates >MBQL (0.09 pmol $CH_4$ $cm^{-3}$ $d^{-1}$ and 0.13 pmol $SO_4^{2-}$ $cm^{-3}$ $d^{-1}$). Red shaded areas in (**e**–**g**) indicate the calculated rates necessary to repair thermal damage (Fig. S3). Data presented in panel **a**–**e** were discussed in a different context by Heuer et al.[12]. Source data for data in panel **e**–**g** are provided as a Source Data file.

abundance from >10,000 to <500 cells $cm^{-3}$ (Fig. 1d), which is interpreted to mark the temperature limit for growth of mesophilic microorganisms[12], and suggests that only a (hyper-) thermophilic subset of the buried microbial community survived the temperature increase above 50 °C.

In sediment samples from <500 mbsf, acetate and methane addition resulted in a strong increase (up to >1000 times) in potential sulfate reduction rates compared to incubations with only trace $H_2$ (Fig. 1f, g). This effect, however, did not occur in the deeper sediment, where the high acetate concentrations (>0.25 mmol $L^{-1}$ below 500 mbsf; Fig. 1b) likely relieved electron donor limitation and substrate competition between sulfate-reducing and methanogenic microbes. We were unable to determine methanogenic activity via $^{14}$C-acetate as the high concentration of unlabeled acetate in the sediment would dilute the radiotracer signal and raise the detection limit above the in-situ rates. However, in marine sediment, especially at high temperatures, methanogenesis from acetate is expected to proceed mainly via a syntrophic electron transfer, whereby acetate is first oxidized to $CO_2$ while the electrons feed autotrophic methanogens that reduce $CO_2$[15,24,25]. We, therefore, suggest that methanogenesis rates based on $^{14}$C-DIC labeling are representative of the overall methanogenic potential (including acetate consumption) in the sediment.

**Potential sediment carbon and biomass turnover**. We tested the plausibility that potential rates determined in the deep sediment represent in-situ values, by comparing these rates to the depletion of dissolved sulfate and to the availability of organic carbon in the sediment. Below 350 mbsf, experimental sulfate reduction and methanogenesis rates dropped to ~0.3 pmol ($CH_4$ or $SO_4^{2-}$) $cm^{-3}$ $d^{-1}$. Assuming that sulfate started at seawater concentration (~28 mmol $L^{-1}$), it would

take ~90,000 years to completely deplete sulfate at this rate (see Methods). The actual sulfate depletion is close to this, between 22 and 28 mol $L^{-1}$, and corresponds to the oxidation of about 0.0084 to 0.011 wt% $C_{org}$, which is only a small fraction of the $C_{org}$ remaining in the bottom ~450 m of the sediment column (0.2–0.4 wt%). We argue that sulfate depletion is likely to have started following heating related to the rapid burial ~400,000 years ago, since non-heated sites with similar $C_{org}$ and age, such as ODP Site 1225[26], have sulfate depletions of less than a few mmol $L^{-1}$. The calculated in-situ depletion reflects the time and spatially averaged depletion rate. Thus, considering that it takes time for rates to increase following burial heating and that there is spatial variation in cell abundances, there is good agreement between the two methods.

We calculated potential biomass turnover times based on our metabolic rate and cell count data, assuming an average carbon content of 22 fg biomass-C per cell[27]. The growth yield, i.e., the efficiency of carbon assimilation, in marine sediment is estimated to range between 0.08 and 0.2 (C-mol biomass/C-mol substrate)[28–30], except for environments where the available energy flux is not sufficient for the synthesis of new biomass[31]. Direct measurements in ocean water revealed microbial growth yields as low as 0.01[32]. With the most conservative assumptions that all detected cells were sulfate reducers and/or methanogens with a carbon assimilation efficiency of 0.01, we calculate potential biomass turnover times for sediment at Site C0023 that range from days to a few years (Fig. 2). Actual biomass turnover is likely much faster, e.g., 10-times faster for a microbial population with 10% sulfate reducers and methanogens or 20-times faster with a growth yield of 0.2. Thus, biomass turnover in the deep, hot sediment at Site C0023 must be many orders of magnitude faster than in colder deep sediment, where microbial biomass turnover times reach hundreds to thousands of years[33]. These estimates for the deep, hot sediment (>40 °C)

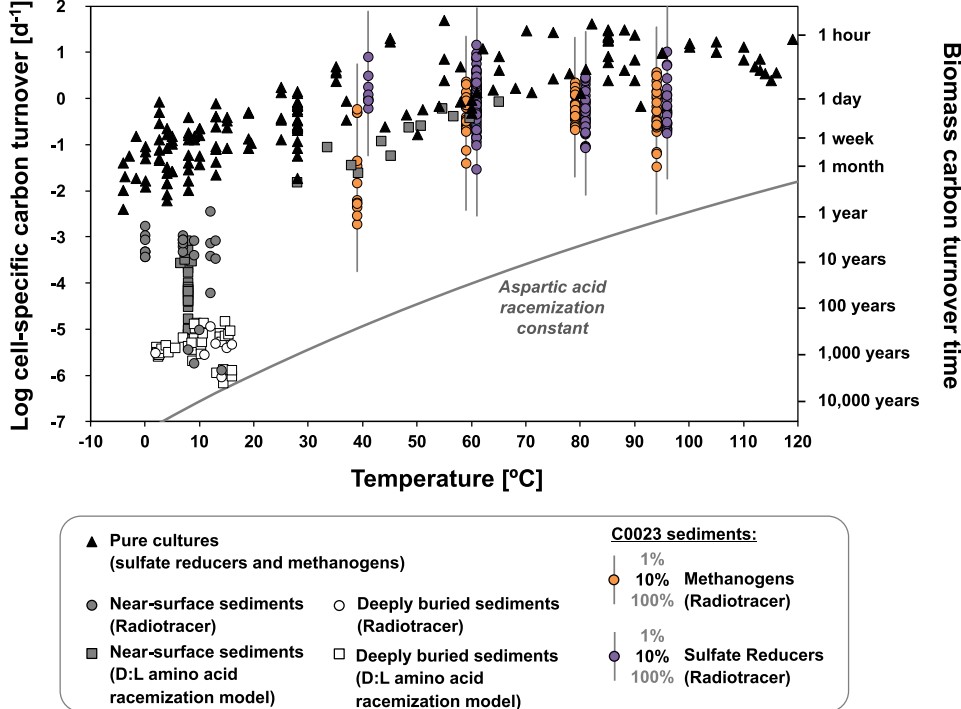

**Fig. 2 Estimates of potential cell-specific carbon turnover in C0023 sediment in relation to temperature, and comparison with microbial turnover in other marine sediment[33,34,67–71,73–75] and in pure cultures[11,38–42,70].** Data were compiled and recalculated as mol $C_{assimilated}$ per mol $C_{cell}$ per day (see Supplementary Material). For the pure cultures, this turnover corresponds to generation times. Bars indicate the range of potential C0023 biomass turnover estimates for a prokaryotic population with 1–100% sulfate reducers and/or methanogens. As acetate and methane additions resulted in a stimulation of sulfate reduction in <350 mbsf sediments, estimates of potential biomass turnover time in these depths are solely based on activity measurements from incubations with trace $H_2$. D:L = ratio between the D and L isomeric forms of aspartic acid. Symbols referring to data sets from different marine sediments, cultures, and incubations are defined in the legend. Source data are provided as a Source Data file.

are consistent with amino acid racemization-based models of fast microbial turnover (days to months) in hydrothermal surface sediment[34].

**Rapid metabolism and thermodynamic constraints to habitability in the deep, hot subseafloor biosphere.** If in-situ rates of sulfate reduction and methanogenesis were on average indeed >0.1 pmol ($CH_4$ or $SO_4^{2-}$) $cm^{-3}$ $d^{-1}$ below 350 mbsf, then an observed microbial community of less than 500 cells per $cm^3$ (Fig. 1d) could have maintained average cell-specific metabolic rates of >0.2 fmol ($CH_4$ or $SO_4^{2-}$) $cell^{-1}$ $d^{-1}$. This calculation, however, assumes that all reported cells are sulfate reducers and/or methanogens. If their respective fraction is lower, then cell-specific rates would be higher. For example, if sulfate reducers and/or methanogens represent <10% of the microbial community, as reported in marine sediment from shallower depths with in-situ temperatures well below 45 °C[25,33], then cell-specific metabolic rates would have been >2 fmol ($CH_4$ or $SO_4^{2-}$) $cell^{-1}$ $d^{-1}$. As all attempts to sequence the DNA of the indigenous deep microbes have failed, because of extremely low cell abundances, the microbial community composition in Site C0023 sediment remains unknown[12,18]. Due to the exceptionally high porewater acetate concentrations, the majority of sulfate reducers and syntrophic methanogens are likely to use acetate as the main substrate. Irrespective of the relative abundance of sulfate reducers and/or methanogens, potential cell-specific rates in this deep, hot sedimentary habitat are high for the deep subseafloor biosphere and are closer to rates found in active surface sediment, rather than in energy-poor deep sediment[33,35]. Methanogenic and sulfate-reducing microorganisms in cold sediment (<15 °C) generally operate at cell-

specific metabolic rates of <1 fmol ($SO_4^{2-}$ or $CH_4$) $cell^{-1}$ $d^{-1}$ in the upper sediment[16,25,33,36]. The cell-specific rates of sulfate-reducing microorganisms may decrease to <0.01 fmol $SO_4^{2-}$ $cell^{-1}$ $d^{-1}$ below 1 mbsf, and to <0.001 fmol $SO_4^{2-}$ $cell^{-1}$ $d^{-1}$ in the deep subseafloor[16,33,36]. Indeed, incubations of sediment from 2-km deep, 40–60 °C warm coal bed with stable isotope-labeled nutrients suggested average cell-specific metabolic rates of ~0.002 fmol of C $cell^{-1}$ $d^{-1}$, whereby <61% of all cells in the sediment appeared to be active[37]. By contrast, respiration rates for pure cultures of sulfate-reducing or methanogenic microorganisms are typically >1 fmol ($SO_4^{2-}$ or $CH_4$) $cell^{-1}$ $d^{-1}$ at psychrophilic to mesophilic temperatures (<45 °C), and can reach several hundred to thousand fmol ($SO_4^{2-}$ or $CH_4$) $cell^{-1}$ $d^{-1}$ for (hyper-)thermophiles[16,33,38–42]. 

The energy required to repair thermal damage to cellular components increases exponentially with temperature, and most of this energy is likely necessary to counteract the continuous racemization of amino acids (Fig. S3)[5,43]. Recycling of amino acids from necromass, which would be less energy-intensive, seems unlikely, given that extracellular amino acids are rapidly degraded in hot sediment >45 °C[34]. High cellular catabolic activity may therefore indeed be necessary to survive in the deep, hot subseafloor. The mortality of cells increases exponentially with the extent of racemization, and proteins possibly have to be replaced when 3% or less of the amino acids within the protein have been racemized[44]. Thus, at 100 °C (920 mbsf), we estimate that the energy turnover required for microorganisms to counteract thermal damage in proteins is >$10^{-14}$ kJ $cell^{-1}$ $d^{-1}$ (Fig. S3). The available Gibbs free energy of sulfate reduction and methanogenesis coupled to the consumption of acetate in Site C0023 sediment increases with depth with a theoretical energy

yield of about $-105\,kJ\,(mol\,SO_4^{2-})^{-1}$ and $-43\,kJ\,(mol\,CH_4)^{-1}$ at 920 mbsf (100 °C). Part of the high energy yield is due to the low dissolved $CO_2$, driven by the large diffusive flux of $Ca^{2+}$ from the basement which leads to carbonate precipitation. From these data, we calculate that sulfate reducers and methanogens would have to maintain cell-specific rates of $>0.1\,fmol\,SO_4^{2-}\,cell^{-1}\,d^{-1}$ and $>0.2\,fmol\,CH_4\,cell^{-1}\,d^{-1}$ to account for the estimated energetic costs of protein repair. The combined energetic costs to repair all vital biomolecules, as well as to maintain cellular integrity and a functioning metabolism at in-situ conditions are likely much higher[5]. Nonetheless, our calculations demonstrate that microorganisms in hot sediment must rely on rapid cellular substrate turnover, which limits the microbial community size that can be maintained, and correspondingly the substrate consumption rate.

Our findings have profound implications for the bounds of life in the deep biosphere. In deep, cold subseafloor sediment, extremely slow growth and strong energy limitation rule microbial subsistence[28,33,45]. By contrast, we now show that in deep, hot subseafloor sediment, the intense in-situ production of organic substrates, such as acetate, can support a minute but highly active population of sulfate-reducing and methanogenic microbes. The high cell-specific catabolic rates result in biomass turnover times of hours to weeks, which is extremely short for the very deep subseafloor biosphere. Rather than merely undergoing selective survival, as is the case for energy-limited communities in cold and temperate sediment[29], rapid biomass turnover and DNA repair could allow cells to evolve and adapt to the extreme conditions in this deep, hot sedimentary habitat. High temperatures result in higher reaction rates, but the lifetimes of complex organic molecules are drastically reduced. As processes of thermal cell damage accelerate above 45 °C, the microorganisms are forced to invest most of their energy metabolism only to maintain cell integrity and cell function. Despite a high activity level, the microbial community is decimated at high temperatures and unable to deplete the available acetate pool. Thus, cells in the deep, hot subseafloor biosphere appear to be balanced on a knife-edge between subsistence near the potential upper-temperature limit for life and a rich surplus of energy and bioavailable organic carbon.

## Methods

**IODP Expedition 370, Site C0023**. IODP Site C0023 is located ~125 km off Cape Muroto, Japan, in the Nankai Trough subduction zone, at the boundary between the younger, hot Philippine Sea plate and the Eurasian plate, (32°22.00′N, 134°57.98′E, 4776 m water depth). Sites 808 and 1174 of the Ocean Drilling Program (ODP)[46] are in the vicinity of IODP Site C0023. More information about the geological setting of Site C0023 is provided in Heuer et al.[18]. Site C0023 was drilled and cored with DV *Chikyu* to a total depth of 1180 m below seafloor (mbsf). The goal of IODP Expedition 370 was to explore the habitability of the deep, hot subseafloor. Site C0023 was chosen due to the high local heat flow leading to a gradual temperature increase (110 °C km$^{-1}$), spanning from 2 °C at the seafloor to $120 \pm 3$ °C in the deepest core retrieved from the basement at 1177 mbsf, which covers the currently known temperature range of microorganisms[18]. Temperature estimates are based on a steady-state temperature profile, that was constrained from lithological properties, thermal conductivity measured in the cores, and in-situ measurements from 189.3 to 407.6 mbsf during drilling[18]. The resulting temperature projections were confirmed by post-cruise monitoring of temperatures down to 860 mbsf in a borehole observatory that was installed after drilling operations concluded[47]. Sediment cores (3–9 m lengths) were recovered by a short-advance, modified hydraulic piston scoring system from 189 to 408 mbsf, and by continuous rotary core barrel coring from >410 mbsf to the bottom of the hole. Sediment includes hemipelagic and pelagic muds and mudstones, turbidite-deposited mud, mudstone, silts, and sands. Lithostratigraphic units were defined on the basis of visual core descriptions: axial trench-wedge facies (Subunit IIA, 189–318.5 mbsf), outer trench-wedge facies (Subunit IIB, 353–428 mbsf), trench-to-basin transitional facies (Subunit IIC, 428–494 mbsf), upper Shikoku Basin facies (Unit III, 494–637.25 mbsf), lower Shikoku Basin facies (Unit IV, 637.25–1112 mbsf), acidic volcaniclastics (Unit V, 1112–1125.9 mbsf), and basaltic basement (Unit VI, 1125.9–1177 mbsf).

**Sampling procedure and quality control**. A wide range of gas, fluid, and solid-phase samples was collected for shipboard and post-cruise research. Rigorous quality assurance and quality control measures were applied during sample recovery and processing to account for multiple sources of contamination, such as the introduction of microbial cells from drilling fluid, from airborne particles, or from reagents or tools used during sample processing. Details on the sampling procedures and contamination control are reported by Heuer et al. 2017 and 2020[12,18] and briefly summarized in the following. Typically, a whole round core (WRC) for gas and solid phase analysis was collected adjacent to the interstitial water (IW) samples and close to microbiology samples (radiotracer incubations and microbial cell counts). Samples for gas analyses were taken from the freshly cut core and transferred into tightly closed glass vials. IW samples were collected by squeezing WRC samples in titanium squeezers. Samples for radiotracer incubations and microbial cell counts were taken from the very center of freshly retrieved, undisturbed core sections. Undisturbed parts of core sections were identified by careful visual inspection and by X-ray computed tomography image scans and transferred to a glove box that was equipped with a tabletop air filtration unit to maintain anaerobic conditions and minimize contamination from airborne particles. Inside the glove box, the outer centimeter of the core was carefully scraped off with sterile ceramic knives. Scraped WRCs were then either directly processed (onboard radiotracer incubations) or packed and flushed with $N_2$ gas into ESCAL gas-barrier bags prior to vacuum-sealing (V-301G, Fuji Impulse, Japan) for transport and microbiological analyses (shore-based radiotracer incubations and microbial cell counts).

**Microbial cell counts**. Details on the cell extraction and counting procedures and corresponding data are reported by Heuer et al. 2017 and 2020[12,18]. All sample preparation for cell counting was carried out in the super-clean room at Kochi Core Center, Japan. Briefly, microbial cells were detached from the sediment matrix by ultrasonication using a detergent mixture (100 mM ethylenediamine-tetraacetic acid, 100 mM sodium pyrophosphate, 1% [v/v] Tween-80), subsequently recovered by density gradient centrifugation and concentrated on polycarbonate membrane filters, as described before[48,49]. Cell concentrates were stained with SYBR Green I, and manually counted under the microscope[50]. At least 900 fields of view were analyzed for each whole membrane.

**Inorganic and organic geochemistry**. Details on the analytical procedures and corresponding data are reported by Heuer et al. 2017 and 2020[12,18] and briefly summarized in the following. For shipboard analysis of methane, gas sample vials were heated to 70 °C for 30 min in an Agilent Technologies 7697A headspace sampler, before an aliquot was injected into an Agilent 7890B gas chromatograph (GC) equipped with an HP PLOT-Q column and flame ionization detector. Shipboard analysis of IW sulfate was conducted using a Dionex ICS-2100 ion chromatograph. Onshore analysis of IW acetate was conducted using isotope ratio monitoring high-performance liquid chromatography/mass spectrometry, as described before[51]. Shipboard analysis of IW DIC was conducted using an automated infrared inorganic carbon analyzer, consisting of a syringe module, a sample-stripping manifold, and an LI-COR LI-7000 $CO_2/H_2O$ analyzer. For shipboard analysis of porewater $H_2$, duplicate 5 cm$^3$ sediment samples from the interior of a WRC were extracted with 18 MΩ water for up to 24 h in closed 20 mL borosilicate glass headspace vials with 1 cm$^3$ $N_2$ headspace[52]. Gas samples from the equilibrated $N_2/H_2$ headspace were analyzed with a PeakPerformer 1 GC. For shipboard analysis of inorganic carbon, powdered sediment samples were reacted with 2 M HCl. The liberated $CO_2$ was then reacted with a monoethanolamine solution, titrated with electrochemically generated $OH^-$ to a colorimetric endpoint, and the resulting change in light transmittance was monitored with a Coulometrics 5012 $CO_2$ coulometer. Shipboard analysis of total carbon was conducted on freeze-dried and powdered sediment samples in tin cups using a Thermo Finnigan Flash EA 1112.

**Radiotracer incubations**. For shipboard determination of rates of methanogenesis and sulfate reduction, 5 mL of sediment from the innermost, undisturbed part of a WRC was placed into a 20 mL crimp vial to which 5 mL of artificial seawater medium was added (recipe below) to facilitate the even distribution of the later added radiotracers. Three replicate vials were prepared from each sample. Vials were crimp-sealed with chlorobutyl stoppers (Bellco) and aluminum crimps. After sealing, the vial headspace was flushed with $N_2$ to remove surplus hydrogen, followed by the addition of 40 µl $N_2/H_2$ gas (95%/5%) to provide approx. 130 nmol L$^{-1}$ of dissolved hydrogen in the liquid phase after equilibration (i.e., close to in-situ $H_2$ concentrations). All vials, stoppers, and equipment that came in contact with the samples prior to or during incubation were autoclaved; solutions were either autoclaved or filtered through sterile syringe filters (0.2 µm pore size) prior to use. Gases were also filtered through sterile syringe filters (0.2 µm pore size) prior to introduction into vials.

The seawater medium for sediment slurry incubations was prepared as follows. The subsequent salts were added to a 2 L glass bottle: 400 mg $KH_2PO_4$, 500 mg $NH_4Cl$, 1 g $MgCl_2 \times 6H_2O$, 1 g KCl, 300 mg $CaCl_2 \times 2H_2O$, 50 g NaCl, and 1.42 g $Na_2SO_4$. Seawater medium for sulfate reduction incubations contained 5 mmol L$^{-1}$ sulfates. No $Na_2SO_4$ salt was added to the medium for methanogenesis incubations.

The bottle was filled up to 2 L with ultrapure $H_2O$. Some drops of Resazurin solution (100 mg Resazurin in 100 mL $H_2O$) were added as an indicator for anoxic, reducing conditions. The bottle was covered (but not completely closed) with a screw cap and autoclaved. After autoclaving, the medium was purged with $N_2$ gas while still hot (>60 °C). During purging, 10 mL of sterile filtered $NaHCO_3$ solution was added to a final concentration of 0.68 mmol $L^{-1}$ in the medium. The pH was adjusted to 7.5 with sterile-filtered 6.5 % HCl or NaOH solution (v/v). The bottle was then closed with a sterile butyl stopper and a screw cap and ca. 3 mL of sterile filtered $Na_2S$ solution (1.2 g $Na_2S$ in 100 mL $H_2O$) were added through the stopper with a syringe to chemically reduce the medium. The reduction was confirmed by the discoloration of the Resazurin solution.

In the radioisotope laboratory on the DV *Chikyu*, 10 μL of radiolabeled ($^{14}C$) $NaHCO_3$ solution (methanogenesis incubations), or 100 μL of $^{35}S$-$Na_2SO_4$ solution (sulfate reduction incubations) containing up to 5 MBq of radioactivity was injected through the rubber stoppers. These radiotracer additions did not change the concentrations of bicarbonate or sulfate detectably. Samples were shaken, and then incubated for a maximum of 10 days at temperatures within the in-situ range: 40 °C for ≤360 mbsf, 60 °C for 405–585 mbsf, 80 °C for 604–775 mbsf, and 95 °C for ≥816 mbsf. The incubation time was chosen as a compromise between sensitivity and available ship time, as incubations had to be terminated before the end of the cruise. Microbial activity was stopped by injecting 500 μL 50% NaOH (w/v) or 5 mL of 20% zinc acetate-solution (w/v) into each vial of the methanogenesis or sulfate reduction incubations, respectively, and vials were shaken vigorously. Sulfate reduction samples were frozen (−20 °C) until analyses. Samples were shipped either to Aarhus University, Denmark, or GFZ Potsdam, Germany, for analysis of methanogenesis and sulfate reduction rates, respectively.

Three additional sulfate reduction incubation experiments were conducted at GFZ Potsdam, Germany, with sediment samples that were shipped and stored under a $N_2$ atmosphere at 4 °C in sealed, gas-tight aluminum bags under an $N_2$ atmosphere. The effects of storage for several weeks on subsequent microbiological analysis under these conditions are expected to be minimal, even for deeply buried sediment[53]. One experiment was a repetition of the onboard determination for samples that did not contain sufficient radioactivity to detect ultra-low rates of sulfate reduction. In the other two experiments, excess electron donors were added to relieve a potential electron donor limitation (either 5 mmol $L^{-1}$ acetates in the medium, or 100 vol% $CH_4$ in the headspace). Except for the electron donor addition, samples were processed identically to shipboard samples.

Separate types of blank and control samples were produced to account for any contamination and/or non-biological radiotracer turnover (see Fig. S1 and Table S1 for all controls and blanks). Sediment controls were incubated without radiotracer addition for both methanogenesis and sulfate reduction. The radiotracer was added after microbial activity was stopped to check for reactions past incubation. Additionally, medium controls (5 mL sterile medium, no sediment) and drill fluid controls (5 ml of drill fluid from the gap between drill core and liner, taken immediately after core retrieval from ~1110 mbsf) were prepared and incubated at the four temperatures with radiotracer addition to check for reactions in the medium and in contaminating drill fluids, respectively. Potential abiotic tracer turnover by the sediment was quantified in killed controls that were prepared after the cruise by γ-irradiating a subset of samples with 18 kGy and subsequently processing them like regular samples. In total over 700 samples and controls were processed.

**Sulfate reduction rate analysis**. Zinc acetate-fixed sediment slurries from stopped sulfate reduction incubations were quantitatively transferred to 50 mL falcon tubes. Before analysis, samples were centrifuged (10 min at 2500 × g), and a 50 μL sub-sample of the supernatant was added to 4 mL of scintillation cocktail (Ultima Gold, Perkin Elmer) to measure $^{35}S$-sulfate radioactivity. The supernatant was carefully decanted off, and total reduced inorganic sulfur (TRIS = $H_2S$, $S^0$, FeS, and $FeS_2$) in the centrifuged sediment pellet was extracted by a single-step cold chromium distillation[54]. The extracted TRIS, trapped in 7 mL of 5% (w/v) ZnAc-solution, was mixed with an 8 mL scintillation cocktail (Ultima Gold, Perkin Elmer). Radioactivity of $^{35}S$-sulfate and $^{35}S$-TRIS were counted on a TriCarb 2500TR liquid scintillation analyzer (Packard Instrument Company). Sulfate reduction rates (SRR) were calculated as described before[54] (Eq. 1)

$$\text{SRR} = (A_{TRIS}/[A_{TRIS} + A_{SO_4^{2-}}]) \times [SO_4^{2-}] \times 1.06 \times \rho/(t \times m)$$

where $A_{TRIS}$ is the radioactivity of TRIS at the end of the incubation, $A_{SO_4^{2-}}$ is the radioactivity of $SO_4^{2-}$ at the end of the incubation, $[SO_4^{2-}]$ is the total $SO_4^{2-}$ in the sample medium based on the $SO_4^{2-}$ concentration in the medium (5 mM) and in the natural sediment porewater, 1.06 is a correction factor for the expected isotopic fractionation, $\rho$ is the bulk sediment density, $t$ is the incubation time, and $m$ is the sediment mass.

Due to the extremely low turnover in many of the samples, rate measurements require careful evaluation of blank and control measurements as they will directly influence the minimum detection limit (MDL). The MDL is controlled by the counter blank, which is the count rate of the scintillation counter inherent to the environment (electrical noise, ambient γ, cosmic radiation) and scintillation fluid in the absence of any added radioactivity. In our case, counter blanks were quantified by measuring samples that contained the same volumes of scintillation cocktail and 5% (w/v) ZnAc as a sample from a distillation trap. As the counter

blanks vary between experiments, they were quantified separately for each experiment (Table S1, Fig. S1). The sample blank comprises the counter blank, the "distillation" blank, resulting from the distillation equipment (i.e., carryover of radioactivity from one distillation to the next), and the tracer blank, resulting from any kind of non-biogenic sulfate reduction that leads to a transfer of radioactivity from the sample into the trap. To determine the distillation blanks, we carried out control distillations without any sample or radioactivity. In all cases distillation blanks were within one standard deviation of the counter blanks, indicating that no detectable carry-over from one distillation to the next took place (Table S1).

Medium control (sterile medium, no sediment) measurements were consistently carried out with every set of samples (including the irradiated sediment controls), and all other controls (drill fluid, sediment, and killed) were within one standard deviation of their respective medium controls. There is some debate on how quickly sulfate-reducing activity stops after suspension of the sample in ZnAc, especially in consolidated samples like the ones used in this study[55]. We consider the killed controls to be the best option to assess true non-biological turnover and calculated a minimum biological quantification limit of 0.13 pmol $SO_4^{2-}$ $cm^{-3}$ $d^{-1}$ based on the average activity measured in the killed control incubations plus 3 times standard deviation (Fig. S1; Table S1).

**Methanogenesis rate analysis**. $^{14}CH_4$ in stopped methanogenesis incubations was determined by flushing the headspace of each sample vial with $CO_2$-free air at 25 mL $min^{-1}$ for 20 min. $^{14}CH_4$ in the gas stream was oxidized to $^{14}CO_2$ in a quartz glass tube containing CuO pellets, heated to 900 °C. $^{14}CO_2$ from the oven exhaust gas was trapped in 5 mL Carbosorb (Perkin Elmer). The entire gas line was made of glass, which does not absorb $CO_2$, and the gas stream was bubbled through 1 M NaOH before combustion to prevent trace amounts of labeled DIC to penetrate into the oven. The efficiency of methane oxidation was tested by adding known amounts of $CH_4$ to a reaction vessel and following its conversion to $CO_2$ in the exhaust gas. For this, 500 μL of the exhaust gas was regularly injected into a GC equipped with a 0.9 m packed silica gel column of 3.1 mm inner diameter and a flame ionization detector (SRI 310C, SRI Instruments). Conversion efficiencies were always >99%.

After extraction of $CH_4$ from the headspace, a subsample of sediment slurry (100–250 μL) was transferred into a new glass vial, crimp capped with butyl rubber stoppers, and acidified with 2 mL of HCl (6 M) to determine the [$^{14}C$]-DIC in the sample. All $^{14}CO_2$ produced was flushed out of the vial headspace with $N_2$ at 25 mL $min^{-1}$ for 35 min and trapped in 5 mL Carbosorb. The radioactivity of $^{14}CO_2$ was counted in 5 mL scintillation cocktail (Permafluor, PerkinElmer) on a TriCarb 2900TR liquid scintillation analyzer (PerkinElmer). Rates of methanogenesis from DIC ($MGR_{DIC}$) were calculated as described before[15] (Eq. 2)

$$MGR_{DIC} = (A_{CH_4}/[A_{CH_4} + A_{DIC}]) \times [DIC] \times 1.08 \times \rho/(t \times m)$$

where $A_{CH_4}$ is the radioactivity of $CH_4$ at the end of the incubation, $A_{DIC}$ is the radioactivity of DIC at the end of the incubation, [DIC] is the total DIC in the sample medium based on the DIC concentration in the medium (0.677 mM) and in the natural sediment porewater, 1.08 is the correction factor for the expected isotopic fractionation between $^{14}C$ and $^{12}C$[56], $\rho$ is the bulk sediment density, $t$ is the incubation time, and m is the sediment mass. A minimum biological quantification limit (MBQL) of 0.09 pmol $CH_4$ $cm^{-3}$ $d^{-1}$ was calculated from the average activity measured in the killed control ($c_I$) incubations plus 3 times standard deviation (Fig. S1, Table S1).

*Calculation of pore water sulfate utilization and sediment organic carbon oxidation.* At Site C0023 a ~600 m thick succession of up to 16 million-year-old hemipelagic mudstones and tuffs has been buried by an equally thick layer of trench deposits within the past 400,000 years[18]. Here, we show that during this rapid burial and heating period, sulfate concentrations in the bottom ~450 m of the column (>750 mbsf) would be minimally affected by diffusion into the section above 600 m following the burial, and then detail how we calculate sulfate depletion over time based on the experimentally determined rates. We investigate the maximum impact of diffusion on sulfate concentrations using the solution to the time-dependent diffusion equation (Eq. 3)

$$C(z, t) = C(z, 0)\text{erf}\left(z/\left[(2 \times D \times t)/\theta^2\right]^{0.50}\right)$$

where $z$ is depth below 600 m and $t$ is the time since burial onset 400,000 years ago, assuming a constant diffusion coefficient, $D$ (0.054 $m^2$ $year^{-1}$), and tortuosity, $\theta^2$ (4.9), based on the average values for the section[18,57]. Further, we assume a constant initial concentration, $C(z, t = 0)$, a zero concentration boundary at 600 mbsf, $C(z = 600, t = 0)$, and an impermeable bottom boundary at 1200 mbsf. At Site C0023, for the bottom boundary, the choice of Dirichlet or Neumann for the deep boundary condition does not significantly impact the result of Eq. 3 because of the relatively short time-scale of the problem, leading us to use a Neumann bottom boundary condition with its simpler mathematical form for the diffusion equation solution. Sediment $\theta^2$ was calculated from the average measured porosity, $\varphi$ (0.35), and formation factor, $F$ (14),

according to (Eq. 4)[58,59]

$$\theta^2 = \varphi \times F$$

Based on Eq. 3, diffusion into the overlying sulfate depleted section would have, at most, only lowered the sulfate concentration (relative to the concentration at the time of burial by the trench deposits) at 750 mbsf by ~11%, at 800 mbsf by ~3% and deeper than 850 mbsf by less than 1%. We argue that at the start of the rapid burial interval, 400,000 years ago, the section had an average sulfate concentration similar to that of seawater (28 mmol L$^{-1}$). This assumption is based on the observation that sites with similar $C_{org}$ and age, such as ODP Site 1225, have depletions of less than a few mmol L$^{-1}$ SO$_4^{2-}$, and less than a few wt% $C_{org}$[26]. As this maximum depletion due to diffusive loss are small compared to the observed depletions relative to seawater, we can calculate the time it would take to deplete sulfate at the experimentally determined potential sulfate reduction rates (SRR). This depletion time, $\tau$, can be compared to the time since rapid burial heating is a test of the plausibility of the measured potential rates. The calculation of $\tau$ requires the assumption of an initial concentration ([SO$_4^{2-}$]$_{initial}$). Thus $\tau$ is calculated as (Eq. 5): $t = ([SO_4^{2-}]_{initial} \times \varphi)/SRR$.

For the measured SRR of ~0.3 pmol SO$_4^{2-}$ cm$^{-3}$ d$^{-1}$, the calculated value of $\tau$ is $3.3 \times 10^7$ days (~90,000 years). Alternatively, Eq. 5 may be rearranged to calculate the potential SRR over any given time period (SRR = ([SO$_4^{2-}$]$_{initial} \times \varphi)/\tau$). For example, over the entire period of 400,000 years, a sulfate depletion of 22–28 mmol L$^{-1}$ corresponds to an average SRR of 0.05–0.07 pmol SO$_4^{2-}$ cm$^{-3}$ d$^{-1}$. For the conversion of $C_{org}$ with oxidation state 0 (2CH$_2$O + SO$_4^{2-}$ → 2HCO$_3^-$ + HS$^-$ + H$^+$) in sediment with a porosity of 0.35 and a bulk density of 2.2 g cm$^{-3}$, a sulfate depletion between 22 and 28 mmol L$^{-1}$ would correspond to the oxidation of 0.0084 to 0.011 wt% $C_{org}$.

**Thermodynamic calculations, potential biomass turnover, and mean potential cell-specific energy turnover.** The standard Gibbs energy ($\Delta G^0_{insitu}$) of sulfate-dependent AOM (CH$_4$ + SO$_4^{2-}$ → HCO$_3^-$ + HS$^-$ + H$_2$O), sulfate reduction from acetate (SO$_4^{2-}$ + CH$_3$COO$^-$ → HS$^-$ + 2HCO$_3^-$), sulfate reduction from hydrogen (4H$_2$ + SO$_4^{2-}$ + H$^+$ → HS$^-$ + 4H$_2$O), methanogenesis from acetate (CH$_3$COO$^-$ + H$_2$O → CH$_4$ + HCO$_3^-$), and methanogenesis from hydrogen (4H$_2$ + HCO$_3^-$ + H$^+$ → CH$_4$ + 3H$_2$O) was calculated using the SUPCRT/OBIGT software package[60] and reported thermodynamic data[61–63] for in-situ temperatures estimated from the local geothermal gradient[12] (100 °C km$^{-1}$) and a median pressure of 55.8 MPa. The energy of reactions at non-standard conditions ($\Delta G_R$; Fig. S1) was calculated according to (Eq. 6)

$$\Delta G_R = \Delta G^0_{insitu} + R \times T \times \ln Q$$

where $R$ (0.008314 kJ mol$^{-1}$ K$^{-1}$) is the ideal gas constant, $T$ (in K) is the in-situ temperature, and $Q$ denotes the activity quotient of the reactants and reaction products. Activities were estimated by multiplying the measured in-situ concentration of the species with activity coefficients calculated from an extended version of the Debye–Hückel equation[64] for an ionic strength of $I = 0.64$ and in-situ temperatures using the Geochemists Workbench® Software. In depths where HS$^-$ was below detection, we assumed a concentration of 0.1 µmol L$^{-1}$.

$\Delta G_R$ for individual reactions was combined with data from microbial activity measurements (SRR or MGR$_{DIC}$) and cell counts to calculate potential per-cell energy turnover according to (Eq. 7, 8)

[energy turnover(kJ cell$^{-1}$ d$^{-1}$)] = [$\Delta G_R$(kJ mol$^{-1}$ SO$_4^{2-}$ or kJ mol$^{-1}$ CH$_4$)]

$\times$ [SRR(mol SO$_4^{2-}$ cm$^{-3}$ d$^{-1}$) or MGR$_{DIC}$(mol CH$_4$ cm$^{-3}$ d$^{-1}$)]/[cell abundance(cells cm$^{-3}$)]

The metabolic rates necessary to account for thermal cell damage were calculated by comparison of $\Delta G_R$ for individual reactions with the energetic costs for repair of DNA depurination and amino acid racemization, based on published relationships with temperature[5,65], according to (Eq. 9):

[maintenance rate AA + DNA repair(mol CH$_4$ cm$^{-3}$ d$^{-1}$)] = [repair cost(kJ cell$^{-1}$ d$^{-1}$)]

$\times$ [cell abundance (cells cm$^{-3}$)/[$\Delta G_R$(kJ mol$^{-1}$ SO$_4^{2-}$ or kJ mol$^{-1}$ CH$_4$)]

Our DNA and protein repair cost calculations consider genome sizes from 1.6 to 10 megabases and assume that proteins have to be replaced if 3% or less of the amino acids within have been racemized[5,44].

Potential microbial biomass turnover times ($T_b$) were estimated by dividing the microbial biomass present at a given depth by the potential rate of biomass production. The biomass was calculated by multiplying the number of cells per cm$^{-3}$ sediment (>16 cells cm$^{-3}$) with the amount of carbon contained within a single cell, assuming an average cellular carbon content of 22 fg C per cell[27] and a 25% cell recovery from sediment extractions[49,66]. The potential rate of biomass production was calculated from the measured SRR or MGR$_{DIC}$, assuming 2 moles of organic carbon ($C_{org}$) oxidized per mole of sulfate or methane turnover, and a growth yield of at least 1%[28–30,32]. Thus, $T_b$ values were calculated according to (Eq. 10):

$[T_b(d^{-1})] = $ [cell abundance (cells cm$^{-3}$)]

$\times [22 \times 10^{-15}$gC cell$^{-1}$]/[12gC/mol C]/ [SRR(mol SO$_4^{2-}$ cm$^{-3}$d$^{-1}$)

or MGR(mol CH$_4$ cm$^{-3}$d$^{-1}$)] / [2 mol($C_{org}$/mol SO$_4^{2-}$ or CH$_4$

/[0.01 mol C/mol $C_{org}$]

The microbial community composition and the abundance of sulfate reducers and/or methanogens in C0023 sediments remain unknown, and therefore we

discuss $T_b$ for 1 to 100% abundance of sulfate reducers and/or methanogens. For other marine sediments[33,34,67–75], we assumed a 10% abundance of sulfate reducers[33].

**Reporting summary.** Further information on research design is available in the Nature Research Reporting Summary linked to this article.

## Data availability

Shipboard data used in this study are available in the IODP Expedition 370 Proceedings, through the J-CORES database, under accession code http://sio7.jamstec.go.jp/j-cores.data/370/C0023A/). Data presented previously reported by Heuer et al.[12] are available in the PANGAEA database under accession code https://doi.org/10.1594/PANGAEA.923088. New data reported in this manuscript are available in the PANGAEA database under accession code https://doi.org/10.1594/PANGAEA.930403 and provided as a Source Data file with this paper. Figures containing raw data from these databases include Figs. 1, 2 and S1, S3. Source data are provided with this paper.

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

## Acknowledgements

This research used samples and data provided by the International Ocean Discovery Program (IODP). We thank all crews, drilling team members, lab technicians, radio safety officers, and scientists on the DV *Chikyu* for supporting core sampling and onboard measurements. In particular, we would like to thank Hiroyuki Imachi, Lorenzo Lagostina, Kyle Metcalfe, Donald Pan, Maija Raudsepp, and Bernhard Viehweger for their outstanding contribution to the careful processing of cores for rate measurements. We further thank Yusuke Kubo and Lena Maeda for their excellent management of expedition logistics. This is a contribution to the Deep Carbon Observatory (DCO). This work was supported in part by the Japan Society for the Promotion of Science (JSPS) Strategic Fund for Strengthening Leading-Edge Research and Development (to JAM-STEC and F.I.), the JSPS Funding Program for Next Generation World-Leading Researchers (GR102 to F.I.), the Deutsche Forschungsgemeinschaft through projects 387745511 (to V.B.H.), 408249062 (to J.K.), and through the Cluster of Excellence "The Ocean Floor—Earth's Uncharted Interface" (Germany's Excellence Strategy—EXC-2077—390741603; to K.-U.H. and V.B.H.), the IODP U.S. Science Support Program (National Science Foundation prime award OCE-1450528 to T.T.), as well as the Danish National Research Foundation Grant DNRF104 and the FP7 European Research Council (ERC) Advanced Grant 294200 (to B.B.J.). Additional support enabling this project was provided by the Deep Carbon Observatory.

## Author contributions

All co-authors designed the research. F.B. wrote the paper with input and edits from all co-authors. T.T. carried out shipboard incubations. R.R.A., F.B., K.L.H., J.K, S.J.E.K., A.J.S., F.S., J.S., and T.T. carried out shore-based incubations and/or analyses of samples. F.B. and C.G. conducted the thermodynamic calculations. A.J.S. and B.B.J. prepared the sulfate depletion model. K.-U.H., V.B.H., F.I., and Y.M. designed, planned, and/or lead IODP Expedition 370.

## Competing interests

The authors declare no competing interests.
