## [Peer Review File · Nature Communications]

Reviewers' Comments:

Reviewer #1:

Remarks to the Author:

This study investigates microbial life in 'hot' seafloor sediments using laboratory-based radiotracer experiments to determine the rates of three different microbial metabolisms commonly found in sedimentary environments (sulfate reduction, methanogenesis and anaerobic oxidation of methane). The radiotracer experiments demonstrate high rates of microbial activity and biomass turnover estimates are determined to be many orders of magnitude faster than colder deep sediments where cell turnover is thought to occur on the order of hundreds to thousands of years. Further, the authors argue that microbes in these hot sediments rely on rapid cellular substrate turnover which ultimately limits the size of the microbial community that can be maintained in this environment.

Overall, I found the study to be well-thought out, detailed and of high-quality. The results are fascinating and provide substantial new insights into how life thrives in hot subsurface environments. I recommend this manuscript be accepted for Nature Communications as is. It was truly a pleasure to read and I look forward to seeing it in print. My only disappointment is that the authors (or others involved with this cruise?) have been unable to obtain high-quality DNA for genomic sequencing. Given that genomic work has been successfully carried out on other low biomass subsurface samples, I hope the authors can resolve this issue since genomic results would help constrain some their assumptions regarding microbial community composition in these samples.

-Nagissa Mahmoudi

Reviewer #2:

Remarks to the Author:

This manuscript seeks to understand the upper temperature limits to active microbial life in marine sediments. The manuscript builds on prior isotopic work examining biochemistry at this site, and takes it a step further, through directly measuring activity via highly sensitive radiotracer experiments.

The manuscript presents compelling evidence that active microbial life (methanogenesis, sulfate reduction) is present at temperatures above 80 C, even though the sediments are low in biomass. The authors also calculate the energy required to maintain cell integrity at high temperatures, and convincingly argue that microorganisms at such hot temperatures would have to maintain relatively high rates metabolism for survival. This is in stark contrast to other deep subsurface environments, where survival often is maintained at very low energy expenditure. The conclusions of the manuscript are supported by the well thought out and executed experiments.

This manuscript will be of interest to a broad audience of deep subsurface investigators, microbiologists, astrobiologists, as well as readers simply interested in the boundaries microbial life on Earth.

This is certainly the easiest review I have done in my career so far - I recommend accepting (almost) as is. My only suggestions are very minor and purely editorial:

- 1) In the Supplementary section (Line 37) remove "see above" for recipe and replace with reference to main text
- 2) It would be very useful if the concentrations of hydrogen, acetate and methane amendments were be added to the Figure legends of Figure 1 and Figure S2 for ease of reference.

Manuscript: NCOMMS-21-12665

Title: Rapid Metabolism Fosters Microbial Survival in the Deep, Hot, Subseafloor Biosphere

Authors: Beulig et al.

Beulig et al. measured potential $^{35}\text{S-SO}_4^{2-}$ reduction rates and $^{14}\text{C-HCO}_3^-$ reduction rates for hot (40 – 95°C), deeply buried (200 – 1100 mbsf) sediments from the Nakai Trough subduction zone. Their experimental protocol included a comprehensive set of controls to account for any radiochemical signals produced by abiotic processes. The results provide clear evidence of biological sulfate reduction and methane production from DIC throughout the sediment column over the full temperature range.

The authors note that their measured sulfate-reduction and methane-production rates are “potential rates” since the manipulations involved in conducting the experiments (e.g., diluting sediment with artificial seawater and supplementing the slurries with trace H_2 [130 nM] and sulfate, and in some experiments acetate and methane) could stimulate or inhibit rates compared to *in situ*. The authors test the plausibility that their potential sulfate reduction rates determined in the deep sediment ($\sim 0.3 \text{ pmol} \cdot \text{cm}_{\text{ws}}^{-3} \cdot \text{d}^{-1}$) represent *in situ* values by calculating that it would take 94,000 years for sulfate to be fully depleted from an initial value of 28 mM. This time for depletion is considered to be in remarkably good agreement with the onset of rapid burial and heating of the bottom 600 m of the sediment column, which began 400,000 years ago.

A more direct approach to constrain *in situ* sulfate-reduction rates (*SRR*) consistent with the geochemical data is to simply calculate the average rate needed to deplete SO_4^{2-} to 6 mM (800 – 1100 mbsf) in 400,000 years:

$$SRR \sim \frac{22 \text{ } \mu\text{mol}}{\text{cm}_{\text{iw}}^3} \cdot \frac{0.35 \text{ cm}_{\text{pw}}^3}{\text{cm}_{\text{ws}}^3} \cdot \frac{10^6 \text{ pmol}}{\mu\text{mol}} \cdot \frac{1}{400,000 \text{ y}} \cdot \frac{\text{y}}{365 \text{ d}} = 0.05 \text{ pmol} \cdot \text{cm}_{\text{ws}}^{-3} \cdot \text{d}^{-1}.$$

This estimate of the *in situ* *SRR* looks to be about an order-of-magnitude below the average potential sulfate reduction rate measured in the 800 – 1100 mbsf interval. However, this rate is a lower-limit since it assumes that sulfate reduction commences relatively quickly (within, say 10^5 y) following the period of rapid burial and heating of the deep sediment.

A similar approach may be used to estimate *in situ* methane-production rates (*MPR*). Methane addition does not stimulate potential sulfate reduction rates at sediment depths >600 mbsf, suggesting that methane oxidation is not a major component of the potential sulfate reduction rates. (Potential methane oxidation rates measured at this site are equivocal owing to abiotic conversion of $^{14}\text{C-CH}_4$ to $^{14}\text{C-CO}_2$ and loss of headspace CH_4 during sample processing.) The average rate needed to produce $\sim 1 \text{ mM CH}_4$ (800 – 1100 mbsf) following rapid burial of the lower 600 m of sediment is:

$$MPR \sim \frac{1 \text{ } \mu\text{mol}}{\text{cm}_{\text{iw}}^3} \cdot \frac{0.35 \text{ cm}_{\text{pw}}^3}{\text{cm}_{\text{ws}}^3} \cdot \frac{10^6 \text{ pmol}}{\mu\text{mol}} \cdot \frac{1}{400,000 \text{ y}} \cdot \frac{\text{y}}{365 \text{ d}} = 0.002 \text{ pmol} \cdot \text{cm}_{\text{ws}}^{-3} \cdot \text{d}^{-1}.$$

Methane at depths >600 mbsf is thermogenic; hence this rate reflects abiotic methane production. For comparison, potential methane production rates are $>0.1 \text{ pmol} \cdot \text{cm}_{\text{ws}}^{-3} \cdot \text{d}^{-1}$ at sediment depths >350 mbsf.

Another approach to constrain *in situ* sulfate-reduction rates is to use a diffusion-reaction equation to calculate rates consistent with the geochemical data:

$$\frac{\partial c}{\partial t} = \frac{D}{\phi F} \frac{\partial^2 c}{\partial x^2} - \frac{SRR_{ws}}{\phi}$$

Assuming that the section of buried sediment had an initial sulfate concentration of 28 mM at the time of burial, and allowing a diffusive flux into the underlying oceanic basement rock to account for the observed sulfate concentration gradient in the bottom 200 m of sediment, the prominent features in the sulfate profile can be reproduced with an *in situ* sulfate reduction rate of $0.04 \text{ pmol} \cdot \text{cm}_{ws}^{-3} \cdot \text{d}^{-1}$.

The possibility that potential sulfate-reduction and methane-production rates may overestimate *in situ* rates does not negate the main conclusion of this paper: biological sulfate reduction and methane production from HCO_3^- are active in the deep, hot, subseafloor biosphere. However, the biomass turnover times calculated using potential rates (Fig. 2) could overestimate *in situ* turnover times by one (sulfate reducers) or two (methanogens) orders-of-magnitude. Likewise, cell-specific metabolic rates based on potential rates ($>5 \text{ fmol} \cdot \text{cell}^{-1} \cdot \text{d}^{-1}$, assuming sulfate reducers and methanogens represent 10% of the microbial community) could be overestimates. For example, assuming a measured cell concentration of 500 cm^{-3} , 10% sulfate reducers, 25% cell recovery, and $SRR \sim 0.04 \text{ pmol} \cdot \text{cm}_{ws}^{-3} \cdot \text{d}^{-1}$, the cell-specific sulfate reduction rate is $0.2 \text{ fmol} \cdot \text{cell}^{-1} \cdot \text{d}^{-1}$, comparable to rates found in cold surface sediment but far below rates observed for laboratory cultures. Biomass turnover times based on potential rates should be called “potential”. I suggest using the term “potential” to describe biomass turnover times and cell-specific metabolic rates. For example, the title of the paper should read: “Rapid Potential Rates of Metabolism May Foster Microbial Survival in the Deep, Hot, Subseafloor Biosphere”.

Specific comments/suggestions/questions:

Line 99. Insert “potential activity of” in front of “anaerobic oxidation of methane” to note that the experimental design yielded $^{14}\text{C-CH}_4$ cpm but did not allow for calculation of potential rates.

Line 113. The sulfate-reduction and methane-production rates in the killed controls (plus $3\times$ the standard deviation) are reported as 0.13 and 0.09 $\text{pmol}\cdot\text{cm}^{-3}\cdot\text{d}^{-1}$, respectively. I tried to reproduce these values based on $A_{\text{TRIS}} \sim 10$ cpm and $A_{\text{CH}_4} < 10$ cpm for killed controls (Fig. S1(a)) and $A_{\text{SO}_4} = A_{\text{DIC}} \sim 5$ MBq (line 312) = 3×10^8 cpm (the radioactivity of sulfate and DIC at the end of the incubation should be the same as at the start since only a small fraction of reactant pool turned over during the incubation):

$$SRR = \phi[\text{SO}_4^{2-}] \frac{A_{\text{TRIS}}}{A_{\text{TRIS}} + A_{\text{SO}_4^{2-}}} \frac{1.06}{t} = \frac{0.35 \text{ cm}_w^3}{\text{cm}_w^3} \cdot \frac{\sim 5 \text{ } \mu\text{mol SO}_4^{2-}}{\text{cm}_w^3} \cdot \frac{10^6 \text{ pmol}}{\mu\text{mol}} \cdot \frac{10 \text{ cpm}}{(10 + 3 \times 10^8) \text{ cpm}} \cdot \frac{1.06}{10 \text{ d}} = \frac{0.006 \text{ pmol}}{\text{cm}_w^3 \cdot \text{d}}$$

$$MPR = \phi_{\text{DIC}} \frac{A_{\text{CH}_4}}{A_{\text{CH}_4} + A_{\text{DIC}}} \frac{1.08}{t} = \frac{0.35 \text{ cm}_w^3}{\text{cm}_w^3} \cdot \frac{\sim 0.68 \text{ } \mu\text{mol DIC}}{\text{cm}_w^3} \cdot \frac{10^6 \text{ pmol}}{\mu\text{mol}} \cdot \frac{10 \text{ cpm}}{(10 + 3 \times 10^8) \text{ cpm}} \cdot \frac{1.08}{10 \text{ d}} = \frac{0.0009 \text{ pmol}}{\text{cm}_w^3 \cdot \text{d}}$$

The standard deviations for the two killed controls reported in Table S1 are 15-20% and can't account for the difference between reported rates and those calculated above. The text should be clarified to make it easier for the reader to reproduce the reported rates for killed controls.

Line 136. The potential sulfate-reduction and methanogenesis rates used to calculate the sulfate depletion time of 94,000 years in Materials and Methods is 0.3 rather than 0.4 $\text{pmol}\cdot\text{SO}_4^{2-} \cdot \text{cm}^{-3}\cdot\text{d}^{-1}$ (line 424).

Line 138. Change “*Suppl. Material*” to “Materials and Methods”.

Line 249. Does extracting interstitial water by squeezing affect acetate concentrations?

Line 266. Add a statement to the effect that cell extraction efficiency for this site is 25%. (see line 457)

Line 310. NaHCO_3 is not charged.

Line 316. Repeated phrase: “as incubations”.

Line 350 and 394. Interstitial water concentration \times bulk sediment density/sediment mass results in unusual units: $\mu\text{mol} \cdot \text{cm}_w^{-3} \cdot \text{g}_{\text{ws}} \cdot \text{cm}_w^{-3} \cdot \text{g}_{\text{ws}}^{-1} = \mu\text{mol} \cdot \text{cm}_w^{-3} \cdot \text{cm}_w^{-3}$.

Line 373. Here the minimum biological quantification limit is referred to as “MBQL”. Elsewhere (e.g., line 399, Fig. 1) it is “MQL”.

Line 424. I get $\tau = 89,500$ years:
$$\frac{28 \text{ } \mu\text{mol}}{\text{cm}_w^3} \cdot \frac{0.35 \text{ cm}_w^3}{\text{cm}_w^3} \cdot \frac{10^6 \text{ pmol}}{\mu\text{mol}} \cdot \frac{\text{cm}_w^3 \cdot \text{d}}{0.3 \text{ pmol}} \cdot \frac{\text{y}}{365 \text{ d}}$$

Line 439: Clarify whether activities used to calculate ΔG_R (eq. 6) are based on *in situ* concentrations or concentrations in slurries used to measured potential sulfate-reduction and methane-production rates that may have been amended with trace H_2 , acetate and sulfate).

Line 442. Delete “molar”.

Line 445. I missed where per cell energy turnover (eqs. 7 and 8) is presented in the manuscript.

Fig. 1. I’m having a hard time distinguishing the symbol colors in panels (f) and (g). It looks like the +trace H₂ rates are shown in both (f) and (g). This should be noted in the caption.

Fig. S1(a). I think the vertical axis label (right) should be “³⁵S-TRIS – Blank”.

Fig. S1(b). The potential sulfate-reduction rates in (b) appear to me to be not consistent with Δcpm values for ³⁵S-TRIS shown in (a). For example, for the +acetate incubations, the median sulfate-reduction rate is about 1 pmol·cm⁻³·d⁻¹ and the median Δcpm is about 100 cpm. Therefore,

$$SRR = \phi \left[\text{SO}_4^{2-} \right] \frac{A_{\text{TRIS}}}{A_{\text{TRIS}} + A_{\text{SO}_4^{2-}}} \frac{1.06}{t} = \frac{0.35 \text{ cm}_{\text{iw}}^3}{\text{cm}_{\text{ws}}^3} \cdot \sim 5 \mu\text{mol SO}_4^{2-} \cdot \frac{10^6 \text{ pmol}}{\text{cm}_{\text{iw}}^3} \cdot \frac{100 \text{ cpm}}{\left(100 + 3 \times 10^8\right) \text{cpm}} \cdot \frac{1.06}{10 \text{ d}} = \frac{0.06 \text{ pmol}}{\text{cm}_{\text{ws}}^3 \cdot \text{d}}$$

Table S1. Killed controls for ³⁵S-SO₄²⁻ incubations are reported as “not determined”. However, results from ³⁵S-SO₄²⁻ incubation killed controls are shown in Fig. S1.

Fig. S3 caption, line 100. What is meant by “energy flux”?

Reviewer #1 (Remarks to the Author):

This study investigates microbial life in 'hot' subseafloor sediments using laboratory-based radiotracer experiments to determine the rates of three different microbial metabolisms commonly found in sedimentary environments (sulfate reduction, methanogenesis and anaerobic oxidation of methane). The radiotracer experiments demonstrate high rates of microbial activity and biomass turnover estimates are determined to be many orders of magnitude faster than colder deep sediments where cell turnover is thought to occur on the order of hundreds to thousands of years. Further, the authors argue that microbes in these hot sediments rely on rapid cellular substrate turnover which ultimately limits the size of the microbial community that can be maintained in this environment.

Overall, I found the study to be well-thought out, detailed and of high-quality. The results are fascinating and provide substantial new insights into how life thrives in hot subsurface environments. I recommend this manuscript be accepted for Nature Communications as is. It was truly a pleasure to read and I look forward to seeing it in print. My only disappointment is that the authors (or others involved with this cruise?) have been unable to obtain high-quality DNA for genomic sequencing. Given that genomic work has been successfully carried out on other low biomass subsurface samples, I hope the authors can resolve this issue since genomic results would help constrain some their assumptions regarding microbial community composition in these samples.

-Nagissa Mahmoudi

Authors' Comment: We thank Reviewer #2 for her supportive comments. DNA analyses were performed with deep-biosphere samples from this expedition. Unfortunately, sequencing was unsuccessful given the extremely low cell numbers.

Reviewer #2 (Remarks to the Author):

This manuscript seeks to understand the upper temperature limits to active microbial life in marine sediments. The manuscript builds on prior isotopic work examining biochemistry at this site, and takes it a step further, through directly measuring activity via highly sensitive radiotracer experiments.

The manuscript presents compelling evidence that active microbial life (methanogenesis, sulfate reduction) is present at temperatures above 80 C, even though the sediments are low in biomass. The authors also calculate the energy required to maintain cell integrity at high temperatures, and convincingly argue that microorganisms at such hot temperatures would have to maintain relatively high rates metabolism for survival. This is in stark contrast to other deep subsurface environments, where survival often is maintained at very low energy expenditure. The conclusions of the manuscript are supported by the well thought out and executed experiments.

This manuscript will be of interest to a broad audience of deep subsurface investigators, microbiologists, astrobiologists, as well as readers simply interested in the boundaries microbial life on Earth.

Authors' Comment: We thank Reviewer #2 for her/his positive feedback.

This is certainly the easiest review I have done in my career so far - I recommend accepting (almost) as is. My only suggestions are very minor and purely editorial:

1) In the Supplementary section (Line 37) remove "see above" for recipe and replace with reference to main text

Authors' Comment: Replaced.

2) It would be very useful if the concentrations of hydrogen, acetate and methane amendments were be added to the Figure legends of Figure 1 and Figure S2 for ease of reference.

Authors' Comment: Information was added.

Reviewer #3 (Remarks to the Author):

Beulig et al. measured potential $^{35}\text{S-SO}_4^{2-}$ reduction rates and $^{14}\text{C-HCO}_3^-$ reduction rates for hot (40 – 95°C), deeply buried (200 – 1100 mbsf) sediments from the Nakai Trough subduction zone. Their experimental protocol included a comprehensive set of controls to account for any radiochemical signals produced by abiotic processes. The results provide clear evidence of biological sulfate reduction and methane production from DIC throughout the sediment column over the full temperature range.

The authors note that their measured sulfate-reduction and methane-production rates are “potential rates” since the manipulations involved in conducting the experiments (e.g., diluting sediment with artificial seawater and supplementing the slurries with trace H_2 [130 nM] and sulfate, and in some experiments acetate and methane) could stimulate or inhibit rates compared to *in situ*. The authors test the plausibility that their potential sulfate reduction rates determined in the deep sediment ($\sim 0.3 \text{ pmol} \cdot \text{cm}_{\text{ws}}^{-3} \cdot \text{d}^{-1}$) represent *in situ* values by calculating that it would take 94,000 years for sulfate to be fully depleted from an initial value of 28 mM. This time for depletion is considered to be in remarkably good agreement with the onset of rapid burial and heating of the bottom 600 m of the sediment column, which began 400,000 years ago.

Authors' Comment: We like to thank Reviewer #3 for this positive in-depth review, in particular for the additional calculations to test (and confirm) the overall outcome of this study.

A more direct approach to constrain *in situ* sulfate-reduction rates (*SRR*) consistent with the geochemical data is to simply calculate the average rate needed to deplete SO_4^{2-} to 6 mM (800 – 1100 mbsf) in 400,000 years:

$$SRR \sim \frac{22 \mu\text{mol}}{\text{cm}_{\text{iw}}^3} \cdot \frac{0.35 \text{ cm}_{\text{pw}}^3}{\text{cm}_{\text{ws}}^3} \cdot \frac{10^6 \text{ pmol}}{\mu\text{mol}} \cdot \frac{1}{400,000 \text{ y}} \cdot \frac{\text{y}}{365 \text{ d}} = 0.05 \text{ pmol} \cdot \text{cm}_{\text{ws}}^{-3} \cdot \text{d}^{-1}.$$

This estimate of the *in situ* *SRR* looks to be about an order-of-magnitude below the average potential sulfate reduction rate measured in the 800 – 1100 mbsf interval. However, this rate is a lower-limit since it assumes that sulfate reduction commences relatively quickly (within, say 10^5 y) following the period of rapid burial and heating of the deep sediment.

Authors' Comment: Thank you for applying an alternative calculation path to test our data. We are glad to hear there is no overall disagreement with our calculations and assumptions.

A similar approach may be used to estimate *in situ* methane-production rates (*MPR*). Methane addition does not stimulate potential sulfate reduction rates at sediment depths >600 mbsf, suggesting that methane oxidation is not a major component of the potential sulfate reduction rates. (Potential methane oxidation rates measured at this site are equivocal owing to abiotic conversion of $^{14}\text{C-CH}_4$ to $^{14}\text{C-CO}_2$ and loss of headspace CH_4 during sample processing.) The average rate needed to produce ~1 mM CH_4 (800 – 1100 mbsf) following rapid burial of the lower 600 m of sediment is:

$$MPR \sim \frac{1 \mu\text{mol}}{\text{cm}_{\text{iw}}^3} \cdot \frac{0.35 \text{ cm}_{\text{pw}}^3}{\text{cm}_{\text{ws}}^3} \cdot \frac{10^6 \text{ pmol}}{\mu\text{mol}} \cdot \frac{1}{400,000 \text{ y}} \cdot \frac{\text{y}}{365 \text{ d}} = 0.002 \text{ pmol} \cdot \text{cm}_{\text{ws}}^{-3} \cdot \text{d}^{-1}.$$

Methane at depths >600 mbsf is thermogenic; hence this rate reflects abiotic methane production. For comparison, potential methane production rates are $>0.1 \text{ pmol} \cdot \text{cm}_{\text{ws}}^{-3} \cdot \text{d}^{-1}$ at sediment depths >350 mbsf.

Another approach to constrain *in situ* sulfate-reduction rates is to use a diffusion-reaction equation to calculate rates consistent with the geochemical data:

$$\frac{\partial c}{\partial t} = \frac{D}{\phi F} \frac{\partial^2 c}{\partial x^2} - \frac{SRR_{\text{ws}}}{\phi}.$$

Assuming that the section of buried sediment had an initial sulfate concentration of 28 mM at the time of burial, and allowing a diffusive flux into the underlying oceanic basement rock to account for the observed sulfate concentration gradient in the bottom 200 m of sediment, the prominent features in the sulfate profile can be reproduced with an *in situ* sulfate reduction rate of $0.04 \text{ pmol} \cdot \text{cm}_{\text{ws}}^{-3} \cdot \text{d}^{-1}$.

Authors' Comment: Again, thank you for applying an alternative calculation path to test our data.

The possibility that potential sulfate-reduction and methane-production rates may overestimate *in situ* rates does not negate the main conclusion of this paper: biological sulfate reduction and methane production from HCO_3^- are active in the deep, hot, seafloor biosphere. However, the biomass turnover times calculated using potential rates (Fig. 2) could overestimate *in situ* turnover times by one (sulfate reducers) or two (methanogens) orders-of-magnitude. Likewise, cell-specific metabolic rates based on potential rates ($>5 \text{ fmol}\cdot\text{cell}\cdot\text{d}^{-1}$, assuming sulfate reducers and methanogens represent 10% of the microbial community) could be overestimates. For example, assuming a measured cell concentration of 500 cm^{-3} , 10% sulfate reducers, 25% cell recovery, and $\text{SRR} \sim 0.04 \text{ pmol}\cdot\text{cm}_{\text{ws}}^{-3}\cdot\text{d}^{-1}$, the cell-specific sulfate reduction rate is $0.2 \text{ fmol}\cdot\text{cell}\cdot\text{d}^{-1}$, comparable to rates found in cold surface sediment but far below rates observed for laboratory cultures. Biomass turnover times based on potential rates should be called “potential”. I suggest using the term “potential” to describe biomass turnover times and cell-specific metabolic rates. For example, the title of the paper should read: “Rapid Potential Rates of Metabolism May Foster Microbial Survival in the Deep, Hot, Subseafloor Biosphere”.

Authors' Comment: "Potential" was added to biomass turnover times and cell-specific metabolic rates where directly derived from potential rates.

We respectfully disagree with the suggested title change because the title is already conservatively hypothetically ("may foster"). Further, since potential rates are produced under modified laboratory conditions after sample recovery, they do not foster microbial survival *in situ* (syntax issue).

Specific comments/suggestions/questions:

Line 99. Insert “potential activity of” in front of “anaerobic oxidation of methane” to note that the experimental design yielded $^{14}\text{C}\text{-CH}_4$ cpm but did not allow for calculation of potential rates.

Authors' Comment: Information added.

Line 113. The sulfate-reduction and methane-production rates in the killed controls (plus $3\times$ the standard deviation) are reported as 0.13 and $0.09 \text{ pmol}\cdot\text{cm}^{-3}\cdot\text{d}^{-1}$, respectively. I tried to reproduce these values based on $A_{\text{TRIS}} \sim 10 \text{ cpm}$ and $A_{\text{CH}_4} < 10 \text{ cpm}$ for killed controls (Fig. S1(a)) and $A_{\text{SO}_4} = A_{\text{DIC}} \sim 5 \text{ MBq}$ (line 312) = $3\times 10^8 \text{ cpm}$ (the radioactivity of sulfate and DIC at the end of the incubation should be the same as at the start since only a small fraction of reactant pool turned over during the incubation):

$$\text{SRR} = \phi \left[\text{SO}_4^{2-} \right] \frac{A_{\text{TRIS}}}{A_{\text{TRIS}} + A_{\text{SO}_4^{2-}}} \frac{1.06}{t} = \frac{0.35 \text{ cm}_{\text{ws}}^3}{\text{cm}_{\text{ws}}^3} \cdot \frac{\sim 5 \text{ } \mu\text{mol SO}_4^{2-}}{\text{cm}_{\text{iw}}^3} \cdot \frac{10^6 \text{ pmol}}{\mu\text{mol}} \cdot \frac{10 \text{ cpm}}{(10 + 3 \times 10^8) \text{ cpm}} \cdot \frac{1.06}{10 \text{ d}} = \frac{0.006 \text{ pmol}}{\text{cm}_{\text{ws}}^3 \cdot \text{d}}$$

$$\text{MPR} = \phi \text{DIC} \frac{A_{\text{CH}_4}}{A_{\text{CH}_4} + A_{\text{DIC}}} \frac{1.08}{t} = \frac{0.35 \text{ cm}_{\text{ws}}^3}{\text{cm}_{\text{ws}}^3} \cdot \frac{\sim 0.68 \text{ } \mu\text{mol DIC}}{\text{cm}_{\text{iw}}^3} \cdot \frac{10^6 \text{ pmol}}{\mu\text{mol}} \cdot \frac{10 \text{ cpm}}{(10 + 3 \times 10^8) \text{ cpm}} \cdot \frac{1.08}{10 \text{ d}} = \frac{0.0009 \text{ pmol}}{\text{cm}_{\text{ws}}^3 \cdot \text{d}}$$

The standard deviations for the two killed controls reported in Table S1 are 15-20% and can't account for the difference between reported rates and those calculated above. The text should be clarified to make it easier for the reader to reproduce the reported rates for killed controls.

Authors' Comment: We thank the reviewer for his/her thorough verification of our calculations. Please note that only the volume of the medium was constant for every experiment. Variations in sediment mass, porosity, porewater volume, porewater concentrations, and injected radioactivity have to be taken into account for every individual

sample, and effectively lead to a much higher experimental error which explains the discrepancy between the reviewer's calculations and our reported minimum biological quantification limit. Below is an example for one of our killed control incubations:

$$MGR_{DIC} = \frac{A_{CH_4}}{A_{CH_4} + A_{DIC}} \times \frac{[DIC]}{t_{incubation} \times m_{sediment}} \times \rho_{Bulk, sediment} \times 1.08$$

$$MGR_{DIC} = \frac{A_{CH_4}}{A_{CH_4} + A_{DIC}} \times \frac{(C_{DIC, Medium} \times V_{DIC, Medium}) + (C_{DIC, PoreWater} \times V_{DIC, PoreWater})}{t_{incubation} \times m_{sediment}} \times \rho_{Bulk, sediment} \times 1.08$$

$$MGR_{DIC} = \frac{11 \text{ cpm}}{11 \text{ cpm} + 6.6 \times 10^7 \text{ cpm}} \times \frac{\left(0.68 \frac{\mu\text{mol}_{DIC}}{\text{cm}^3_{\text{Medium}}} \times 5.0 \text{ cm}^3_{\text{Medium}}\right) + \left(5.0 \frac{\mu\text{mol}_{DIC}}{\text{cm}^3_{\text{PoreWater}}} \times 1.28 \text{ cm}^3_{\text{PoreWater}}\right)}{10 \text{ d} \times 7 \text{ g}_{\text{sediment}}} \times 2.1 \frac{\text{g}_{\text{sediment}}}{\text{cm}^3_{\text{sediment}}} \times 1.08 \times 10^6 \frac{\text{pmol}_{DIC}}{\mu\text{mol}_{DIC}}$$

$$MGR_{DIC} = 0.053 \frac{\text{pmol}_{DIC}}{\text{cm}^3_{\text{sediment}} \text{ d}^{-1}}$$

Line 136. The potential sulfate-reduction and methanogenesis rates used to calculate the sulfate depletion time of 94,000 years in Materials and Methods is 0.3 rather than 0.4 $\text{pmol} \cdot \text{SO}_4^{2-} \cdot \text{cm}^{-3} \cdot \text{d}^{-1}$ (line 424).

Authors' Comment: Thank you for identifying this typo. The main text now reports the correct rates, consistent with those reported in the Materials and Methods section (i.e. $\sim 0.3 \text{ pmol} (\text{CH}_4 \text{ or } \text{SO}_4^{2-}) \text{ cm}^{-3} \text{ d}^{-1}$).

Line 138. Change “*Suppl. Material*” to “Materials and Methods”.

Authors' Comment: Changed.

Line 249. Does extracting interstitial water by squeezing affect acetate concentrations?

Authors' Comment: Squeezing artifacts are expected to be negligible since acetate is a small ion and most exchange sites are selective for cations.

Line 266. Add a statement to the effect that cell extraction efficiency for this site is 25%. (see line 457)

Authors' Comment: Thanks for capturing the missing information. We noticed that the information provided in line 457 was incomplete. The cited study quantified the extraction efficiency of viral particles from sediment of Site C0023. Cell extraction efficiency could not be determined, because cell abundance was too low for robust quantification. However, the 25% cell extraction efficiency used for the calculation in line 457 is based on a previous study by Morono et al. 2013, and it consistent with the determined extraction efficiency of viral particles from sediment of Site C0023. We amended the relevant citation:

Morono, Y., Terada, T., Kallmeyer, J. and Inagaki, F., 2013. An improved cell separation technique for marine subsurface sediments: applications for high-throughput analysis using flow cytometry and cell sorting. *Environmental microbiology*, 15(10), pp.2841-2849.

Line 310. NaHCO₃ is not charged.

Authors' Comment: Charge was deleted.

Line 316. Repeated phrase: “as incubations”.

Authors' Comment: Repeated phrase was deleted.

Line 350 and 394. Interstitial water concentration × bulk sediment density / sediment mass results in unusual units: $\mu\text{mol} \cdot \text{cm}_{\text{iw}}^{-3} \cdot \text{g}_{\text{ws}} \cdot \text{cm}_{\text{ws}}^{-3} \cdot \text{g}_{\text{ws}}^{-1} = \mu\text{mol} \cdot \text{cm}_{\text{iw}}^{-3} \cdot \text{cm}_{\text{ws}}^{-3}$.

Authors' Comment: We assume with "interstitial water concentration" the reviewer is referring to porosity. Porosity is unitless: it is defined as pore (or water) volume per sediment volume.

Line 373. Here the minimum biological quantification limit is referred to as “MBQL”. Elsewhere (e.g., line 399, Fig. 1) it is “MQL”.

Authors' Comment: Thanks for noticing. The reviewer is right that in former line 399 the term should also be MBQL. This has been corrected. In other cases (cell counts and AOM) the term should be MQL. Quantification limits for cells and AOM are not considered biological, because there are no abiotic cells and the detection limit of AOM was not determined from killed sediments but from sterile medium, respectively.

Line 424. I get $\tau = 89,500$ years: $\frac{28 \mu\text{mol}}{\text{cm}_{\text{iw}}^3} \cdot \frac{0.35 \text{ cm}_{\text{iw}}^3}{\text{cm}_{\text{ws}}^3} \cdot \frac{10^6 \text{ pmol}}{\mu\text{mol}} \cdot \frac{\text{cm}_{\text{ws}}^3 \cdot \text{d}}{0.3 \text{ pmol}} \cdot \frac{\text{y}}{365 \text{ d}}$

Authors' Comment: Thank you for identifying the error in our calculations. In an earlier version of the manuscript the depletion time τ was calculated with a porosity of 0.37. The calculated depletion time has been corrected accordingly.

Line 439: Clarify whether activities used to calculate ΔG_{R} (eq. 6) are based on *in situ* concentrations or concentrations in slurries used to measured potential sulfate-reduction and methane-production rates that may have been amended with trace H₂, acetate and sulfate).

Authors' Comment: Text has been changed to clarify that activities were calculated from measured *in-situ* concentrations.

Line 442. Delete “molar”.

Authors' Comment: Deleted.

Line 445. I missed where per cell energy turnover (eqs. 7 and 8) is presented in the manuscript.

Authors' Comment: Per cell energy turnover is presented in (former) Line 189 and in Figure S3.

Fig. 1. I'm having a hard time distinguishing the symbol colors in panels (f) and (g). It looks like the +trace H₂ rates are shown in both (f) and (g). This should be noted in the caption.

Authors' Comment: We changed the colors in Fig. 1 and now highlight that “+trace H₂” rates are shown in both panels (f) and (g).

Fig. S1(a). I think the vertical axis label (right) should be “³⁵S-TRIS – Blank”.

Authors' Comment: Thanks for noticing. Axis label has been corrected.

Fig. S1(b). The potential sulfate-reduction rates in (b) appear to me to be not consistent with Δcpm values for ³⁵S-TRIS shown in (a). For example, for the +acetate incubations, the median sulfate-reduction rate is about 1 pmol·cm⁻³·d⁻¹ and the median Δcpm is about 100 cpm. Therefore,

$$SRR = \varphi \left[\text{SO}_4^{2-} \right] \frac{A_{\text{TRIS}}}{A_{\text{TRIS}} + A_{\text{SO}_4^{2-}}} \frac{1.06}{t} = \frac{0.35 \text{ cm}_{\text{iw}}^3}{\text{cm}_{\text{ws}}^3} \cdot \frac{\sim 5 \mu\text{mol SO}_4^{2-}}{\text{cm}_{\text{iw}}^3} \cdot \frac{10^6 \text{ pmol}}{\mu\text{mol}} \cdot \frac{100 \text{ cpm}}{(100 + 3 \times 10^8) \text{ cpm}} \cdot \frac{1.06}{10 \text{ d}} = \frac{0.06 \text{ pmol}}{\text{cm}_{\text{ws}}^3 \cdot \text{d}}$$

Authors' Comment: Again, we thank the reviewer for his/her thorough verification of our calculations. As explained in previous comment, such discrepancy is the result of variation in experimental parameters (sediment mass, porosity, porewater volume, porewater concentrations, and injected radioactivity).

Table S1. Killed controls for ³⁵S-SO₄²⁻ incubations are reported as “not determined”. However, results from ³⁵S-SO₄²⁻ incubation killed controls are shown in Fig. S1.

Authors' Comment: Killed controls for sulfate reduction are shown in the 6th column from left (third from right). Note that killed controls were only determined during shore-based incubations after the expedition. Table layout was changed to avoid confusion.

Fig. S3 caption, line 100. What is meant by “energy flux”?

Authors' Comment: We believe the reviewer is referring to the typo (“Energety flux”) in Fig. S3 b). The typo was corrected.

Manuscript: NCOMMS-21-12665

Title: Rapid Metabolism Fosters Microbial Survival in the Deep, Hot, Subseafloor Biosphere

Authors: Beulig et al.

The authors misunderstood the main point of my review. I will try to be clearer in my response to their rebuttal. Also, I will share some new insights that I gleaned from reading the revised manuscript.

The authors tested the plausibility that their measured potential sulfate reduction rates ($\sim 0.3 \text{ pmol SO}_4^{2-} \text{ cm}^{-3} \text{ d}^{-1}$ below 350 mbsf) represent *in-situ* values by calculating the sulfate depletion time ($\sim 90,000$ years) and comparing it to the length of time since the sediments were buried ($\sim 400,000$ years). This comparison suggests that the measured potential sulfate reduction rates are plausible if $\sim 300,000$ years had lapsed after sediment burial before the onset of sulfate reduction.

There is a problem with this argument in addition to the lack of evidence presented for a $\sim 300,000$ -year lag period: the sulfate depletion time calculation assumes that the bottom ~ 600 m of the sediment column can be approximated as a closed system. The authors' claim that a diffusion length-scale (L) of 140 m means that "diffusion [over a time period of 400,000 years] would have minimally affected the average sulfate of the lower 600 m of the section". This is not correct. Rather, $L = 140$ m simply means that sulfate ions, on average, diffuse ~ 140 m in 400,000 years. Solution of the diffusion equation ($D = 0.054 \text{ m}^2 \text{ y}^{-1}$, $\theta = 0.35$, and $F = 3$, uniform $[\text{SO}_4^{2-}] = 28 \text{ mM}$ at time = 0, open boundary at the top and closed boundary at the bottom) shows that over a time span of 400,000 years, the bottom ~ 600 m of the sediment column is not well approximated as a closed system. For example, at 140 m (one diffusion length-scale) below the upper boundary, diffusion over a period of 400,000 years reduced the sulfate concentration to about half of its initial value.

I should have better explained in my review that the rationale for using a diffusion-reaction equation is to avoid the closed-system approximation, and thereby improve the reliability of the plausibility test. I showed in my review that allowing for diffusion, the observed sulfate profile below 600 mbsf could be accounted for with an average *in-situ* sulfate reduction rate since burial ($0.04 \text{ pmol cm}_{\text{ws}}^{-3} \text{ d}^{-1}$) that is approximately *one order-of-magnitude lower* than the measured potential sulfate reduction rates.

The same approach applied to potential methane production rates ($\sim 0.3 \text{ pmol CH}_4 \text{ cm}^{-3} \text{ d}^{-1}$ below 350 mbsf) suggested that the average *in-situ* rate since burial is about *two orders-of-magnitude lower* than the measured potential rates. I mentioned in the review that $\delta^{13}\text{C-CH}_4$ values and C_1/C_2 ratios below 800 mbsf indicate that methane in these sediments is thermogenic, consistent with measured potential rates of biogenic methane production that overestimate the *in-situ* rates.

In summary, the authors interpreted the calculations that I did in the review as confirming the plausibility of their measured potential sulfate reduction and methane production rates. Rather, the open system calculation, which I felt improved the plausibility test, was intended to show the opposite: that the measured potential rates may over estimate *in-situ* rates and might best be viewed as upper limits for *in-situ* sulfate reduction and methane production rates.

Why fret over whether *in-situ* sulfate reduction and methane production rates are ~ 0.3 vs. ~ 0.04 and $\sim 0.002 \text{ pmol cm}^{-3} \text{ d}^{-1}$? As I stated in the review, “the possibility that potential sulfate-reduction and methane-production rates may overestimate *in-situ* rates does not negate the main conclusion of this paper: biological sulfate reduction and methane production from HCO_3^- are active in the deep, hot, subsurface biosphere.” The accuracy of the potential rate measurements become important when they are used to calculate cell-specific rates and then compared to cell-specific rates in pure cultures. Given a measured cell density of $\sim 10^2 \text{ cells cm}^{-3}$, 25% cell extraction efficiency, and assuming sulfate reducing and methane producing cells account for 10% of the microbial community, potential sulfate reduction and methane production rates ($0.3 \text{ pmol cm}^{-3} \text{ d}^{-1}$) translate to $7.5 \text{ fmol (CH}_4 \text{ or SO}_4^{2-}) \text{ cell}^{-1} \text{ d}^{-1}$. The authors claim that these cell-specific rates are “similar to rates in laboratory cultures” which are “typically $>1 \text{ fmol (CH}_4 \text{ or SO}_4^{2-}) \text{ cell}^{-1} \text{ d}^{-1}$ ”. This statement is correct if the measured potential rates accurately reflect *in-situ* values. The plausibility test as described in the review suggested that the measured potential rates may overestimate *in-situ* rates (hence cell-specific rates) by one (sulfate reduction) or two (methane production) orders-of-magnitude. This is the reason that I recommended in my review that the authors make clear that their finding — metabolism in the deep, hot subsurface biosphere is unexpectedly rapid — is based on potential rates.

While looking through the revised manuscript, I decided to check some of the references and discovered inconsistencies in how published metabolic rates are reported for colder sediment and pure cultures. The process of reading these papers ultimately led me to realize that metabolism in the deep, hot subsurface biosphere is in fact, unexpectedly *slow*.

The authors write: “Methanogenic and sulfate reducing microorganisms in colder sediment ($<15^\circ\text{C}$) operate at cell-specific metabolic rates of $<0.1 \text{ fmol (CH}_4 \text{ or SO}_4^{2-}) \text{ cell}^{-1} \text{ d}^{-1}$, which decrease steeply to $<0.01 \text{ fmol (CH}_4 \text{ or SO}_4^{2-}) \text{ cell}^{-1} \text{ d}^{-1}$ below 1 mbsf^{31,33}. ... By contrast, respiration rates for pure cultures of sulfate-reducing or methanogenic microorganisms are typically $>1 \text{ fmol (CH}_4 \text{ or SO}_4^{2-})$

cell⁻¹ d⁻¹, and can reach several hundred fmol (CH₄ or SO₄²⁻) cell⁻¹ d⁻¹, especially for (hyper-)thermophiles such as *Archaeoglobus fulgidus* or *Methanocaldococcus str. JH146*^{31,36-38}.”

Reference 31 states that “the *mean* cell-specific rate of sulphate reduction was around 0.1 fmol SO₄²⁻ cell⁻¹ d⁻¹ {italics mine}” (i.e., 0.1 fmol SO₄²⁻ cell⁻¹ d⁻¹ is the average rate rather than an upper limit). Reference 33 states that “sulfate reducers operated at fairly constant metabolic rates of ~0.4 fmol dsrB⁻¹ d⁻¹ throughout the sulfate zone” with no steep decline with depth until sulfate is depleted. Reference 33 also states that below the sulfate depletion depth, “potential *mean* cell-specific methanogenesis rates increased to *average* values of ~0.5 fmol mcrA⁻¹ d⁻¹” {italics mine}; the data show no decline with depth. (Reference 33 assumes that sulfate reducer and methanogen genomes contain a single copy of dsrB or mcrA, respectively.) Reference 36 measured cell-specific sulfate reduction rates for 32 different sulfate reducing prokaryotes; the median value is 17 fmol SO₄²⁻ cell⁻¹ d⁻¹ (range: 0.9-434). Reference 38 reports an average cell-specific methane production rate (apart from super-optimal temperatures) for *Methanocaldococcus str. JH146* of 0.39 pmol cell⁻¹ h⁻¹ (9400 fmol CH₄ cell⁻¹ d⁻¹).

A more accurate representation of these references would be: “Methanogenic and sulfate reducing microorganisms in colder sediment (<15°C) operate at cell-specific metabolic rates of <1 fmol (CH₄ or SO₄²⁻) cell⁻¹ d⁻¹, which may decrease steeply³¹ or remain constant³³ with depth. ... By contrast, the median respiration rate for 32 pure cultures of sulfate-reducing microorganisms is 17 fmol SO₄²⁻ cell⁻¹ d⁻¹ (range: 0.9-434)³⁶ and can reach several hundred fmol SO₄²⁻ cell⁻¹ d⁻¹ for (hyper-)thermophiles such as *Archaeoglobus fulgidus*.³⁷ Respiration rates for (hyper-)thermophilic methanogens such as *Methanocaldococcus str. JH146* can reach nearly ten thousand fmol CH₄ cell⁻¹ d⁻¹.³⁸”

I realize that shifting the rule-of-thumb by an order-of-magnitude (i.e., cold sediments <1 fmol cell⁻¹ d⁻¹, pure cultures typically >~10 fmol cell⁻¹ d⁻¹) does not affect the claim that potential cell-specific sulfate reduction and methane production rates in deep, hot sediments (7.5 fmol cell⁻¹ d⁻¹) are similar to rates in laboratory cultures. However, while reading reference 36 I realized the obvious: temperature must be considered when comparing rates of metabolism. The potential cell-specific rate measured for sulfate-reducers from sub-seafloor sediment incubated at ~80°C with saturating concentrations of acetate is similar to *Desulfofrigus oceanense* (7.6 fmol cell⁻¹ d⁻¹; reference 36) isolated from Arctic sediment and incubated at 9°C with 20 mM acetate. In contrast, *Archaeoglobus fulgidus* isolated from a submarine hot spring and incubated at 80°C with 20 mM lactate has a cell-specific sulfate-reduction rate of 63.8 fmol cell⁻¹ d⁻¹ (reference 36; reference 37 reports a cell-specific sulfate-reduction rate of 336 fmol cell⁻¹ d⁻¹ for the same species grown at 82-84°C). Thus, sulfate reducers in the deep, hot subseafloor biosphere subsist with unexpectedly *low* cell-specific rates of catabolism compared to pure cultures at the same temperature. This provides a simple explanation for the low cell density in these sediments: The power available for growth is equal to the power from the catabolic reaction (rate × ΔG) less the power used for functions other than growth (including maintenance). Since maintenance increases with temperature, cell growth rates (and presumably cell-densities) will decrease with increasing temperature if the rate of catabolism fails to keep up with maintenance. It is important to note that the possibility that metabolism in the deep, hot subsurface biosphere is unexpectedly slow does not diminish the significance of this work.

In conclusion, I hoped to make three points. First, the calculations in my review did not agree with the calculations and assumptions in the manuscript. Second, the characterization of literature values of cell-specific rates is not accurate. (While the errors do not affect the conclusions of this paper, errors in the literature tend to be repeated, and with enough repetition become accepted as fact to the detriment of the scientific community.) Third, temperature must be considered when comparing rates of metabolism.

Response to Reviewer #3

We thank Reviewer #3 for the constructive and helpful comments to our revised manuscript. We highly appreciate the feedback and acknowledge that we misunderstood some of the earlier comments. We are confident that we were able to address all the reviewer's new comments and have implemented important edits to the manuscript. Please find our specific responses below.

Reviewer's Comment:

The authors misunderstood the main point of my review. I will try to be clearer in my response to their rebuttal. Also, I will share some new insights that I gleaned from reading the revised manuscript.

The authors tested the plausibility that their measured potential sulfate reduction rates ($\sim 0.3 \text{ pmol SO}_4^{2-} \text{ cm}^{-3} \text{ d}^{-1}$ below 350 mbsf) represent *in-situ* values by calculating the sulfate depletion time ($\sim 90,000$ years) and comparing it to the length of time since the sediments were buried ($\sim 400,000$ years). This comparison suggests that the measured potential sulfate reduction rates are plausible if $\sim 300,000$ years had lapsed after sediment burial before the onset of sulfate reduction.

There is a problem with this argument in addition to the lack of evidence presented for a $\sim 300,000$ -year lag period: the sulfate depletion time calculation assumes that the bottom ~ 600 m of the sediment column can be approximated as a closed system. The authors' claim that a diffusion length-scale (L) of 140 m means that "diffusion [over a time period of 400,000 years] would have minimally affected the average sulfate of the lower 600 m of the section". This is not correct. Rather, $L = 140$ m simply means that sulfate ions, on average, diffuse ~ 140 m in 400,000 years. Solution of the diffusion equation ($D = 0.054 \text{ m}^2 \text{ y}^{-1}$, $\theta = 0.35$, and $F = 3$, uniform $[\text{SO}_4^{2-}] = 28 \text{ mM}$ at time = 0, open boundary at the top and closed boundary at the bottom) shows that over a time span of 400,000 years, the bottom ~ 600 m of the sediment column is not well approximated as a closed system. For example, at 140 m (one diffusion length-scale) below the upper boundary, diffusion over a period of 400,000 years reduced the sulfate concentration to about half of its initial value.

I should have better explained in my review that the rationale for using a diffusion-reaction equation is to avoid the closed-system approximation, and thereby improve the reliability of the plausibility test. I showed in my review that allowing for diffusion, the observed sulfate profile below 600 mbsf could be accounted for with an average *in-situ* sulfate reduction rate since burial ($0.04 \text{ pmol cm}_{\text{ws}}^{-3} \text{ d}^{-1}$) that is approximately *one order-of-magnitude lower* than the measured potential sulfate reduction rates.

The same approach applied to potential methane production rates ($\sim 0.3 \text{ pmol CH}_4 \text{ cm}^{-3} \text{ d}^{-1}$ below 350 mbsf) suggested that the average *in-situ* rate since burial is about *two orders-of-magnitude lower* than the measured potential rates. I mentioned in the review that $\delta^{13}\text{C-CH}_4$ values and C_1/C_2 ratios below 800 mbsf indicate that methane in these sediments is thermogenic, consistent with measured potential rates of biogenic methane production that overestimate the *in-situ* rates.

In summary, the authors interpreted the calculations that I did in the review as confirming the plausibility of their measured potential sulfate reduction and methane production rates. Rather, the open system calculation, which I felt improved the plausibility test, was intended to show the opposite: that the measured potential rates may over estimate *in-situ* rates and might best be viewed as upper limits for *in-situ* sulfate reduction and methane production rates.

Author's Response:

Reviewer #3 correctly pointed out a problem with the argument we presented. We have taken the reviewer's advice and have now modeled diffusion quantitatively as an open system. Our model results agree with those of the reviewer when we use the formation factor, 3, used by the reviewer. This value for the formation factor, however, which we had in the original paper was unfortunately incorrect (it was an unnormalized resistance value rather than the formation factor which is the resistance normalized to the resistance of seawater). The correct normalized value is 14. This value reduces the depth to which diffusion has significant impact compared to that shown by the reviewer since, to a first order, the depth of the impact of diffusion is inversely related to the square root of the tortuosity.

Based on the reviewer's helpful analysis, we now do not claim that the lower 600 meters can be considered a closed system. Instead, our model result (which is consistent with the reviewer's model) shows that the bottom 450 meters are minimally affected by diffusion into the overlying sediment. This is a more precise statement than we previously used. We have included the analytical solution of the model in the text and have modified the arguments to be consistent with the new result. We find that this modification does not alter our general conclusions, but in fact strengthens them, as they are now based on a more rigorous, quantitative approach. Again, we thank the reviewer for the constructive critique.

Below is a graph of the diffusion solution, as described in detail in the revised Materials and Methods section, below 600 meters at 400,000 years. The vertical axis is the relative starting concentration. The horizontal axis is depth below 600 meters (z in the manuscript). The graph shows how the relative concentration of sulfate changes over 400,000 years by assuming a zero boundary at 600 meters. It is the same curve produced by Reviewer #3 but shifted because of using the correct tortuosity.

Reviewer's Comment:

Why fret over whether *in-situ* sulfate reduction and methane production rates are ~ 0.3 vs. ~ 0.04 and ~ 0.002 $\text{pmol cm}^{-3} \text{d}^{-1}$? As I stated in the review, “the possibility that potential sulfate-reduction and methane-production rates may overestimate *in-situ* rates does not negate the main conclusion of this paper: biological sulfate reduction and methane production from HCO_3^- are active in the deep, hot, subseafloor biosphere.” The accuracy of the potential rate measurements become important when they are used to calculate cell-specific rates and then compared to cell-specific rates in pure cultures. Given a measured cell density of $\sim 10^2$ cells cm^{-3} , 25% cell extraction efficiency, and assuming sulfate reducing and methane producing cells account for 10% of the microbial community, potential sulfate reduction and methane production rates (0.3 $\text{pmol cm}^{-3} \text{d}^{-1}$) translate to 7.5 $\text{fmol} (\text{CH}_4 \text{ or } \text{SO}_4^{2-}) \text{ cell}^{-1} \text{d}^{-1}$. The authors claim that these cell-specific rates are “similar to rates in laboratory cultures” which are “typically >1 $\text{fmol} (\text{CH}_4 \text{ or } \text{SO}_4^{2-}) \text{ cell}^{-1} \text{d}^{-1}$ ”. This statement is correct if the measured potential rates accurately reflect *in-situ* values. The plausibility test as described in the review suggested that the measured potential rates may overestimate *in-situ* rates (hence cell-specific rates) by one (sulfate reduction) or two (methane production) orders-of-magnitude. This is the reason that I recommended in my review that the authors make clear that their finding — metabolism in the deep, hot subsurface biosphere is unexpectedly rapid — is based on potential rates.

Author's Response:

The reviewer is right that accuracy of the potential rate measurements become important when they are used to calculate cell-specific rates and then compared to cell-specific rates in pure cultures. We are confident, given the short incubation time, that a change in rate based on growth can be excluded. Elevated rates could be caused by modified environmental conditions in the slurry, which we made sure is now pointed out clearly in the manuscript. We further cited relevant literature discussing the effect of slurry incubation on metabolic rates [Jørgensen 1978]. Given the two completely different approaches (1-point potential rate measurements vs. a model that spans over 100-thousands of years) as well as the level of uncertainties in both approaches, the numbers still agree sufficiently well to maintain the general conclusion, with a statement about potential rates.

Reviewer's Comment:

While looking through the revised manuscript, I decided to check some of the references and discovered inconsistencies in how published metabolic rates are reported for colder sediment and pure cultures. The process of reading these papers ultimately led me to realize that metabolism in the deep, hot subsurface biosphere is in fact, unexpectedly *slow*.

The authors write: “Methanogenic and sulfate reducing microorganisms in colder sediment (<15°C) operate at cell-specific metabolic rates of <0.1 fmol (CH₄ or SO₄²⁻) cell⁻¹ d⁻¹, which decrease steeply to <0.01 fmol (CH₄ or SO₄²⁻) cell⁻¹ d⁻¹ below 1 mbsf^{31,33}. ... By contrast, respiration rates for pure cultures of sulfate-reducing or methanogenic microorganisms are typically >1 fmol (CH₄ or SO₄²⁻) cell⁻¹ d⁻¹, and can reach several hundred fmol (CH₄ or SO₄²⁻) cell⁻¹ d⁻¹, especially for (hyper-)thermophiles such as *Archaeoglobus fulgidus* or *Methanocaldococcus str. JH146*^{31,36-38}.”

Reference 31 states that “the *mean* cell-specific rate of sulphate reduction was around 0.1 fmol SO₄²⁻ cell⁻¹ d⁻¹ {italics mine}” (i.e., 0.1 fmol SO₄²⁻ cell⁻¹ d⁻¹ is the average rate rather than an upper limit). Reference 33 states that “sulfate reducers operated at fairly constant metabolic rates of ~0.4 fmol dsrB⁻¹ d⁻¹ throughout the sulfate zone” with no steep decline with depth until sulfate is depleted. Reference 33 also states that below the sulfate depletion depth, “potential *mean* cell-specific methanogenesis rates increased to *average* values of ~0.5 fmol mcrA⁻¹ d⁻¹” {italics mine}; the data show no decline with depth. (Reference 33 assumes that sulfate reducer and methanogen genomes contain a single copy of dsrB or mcrA, respectively.) Reference 36 measured cell-specific sulfate reduction rates for 32 different sulfate reducing prokaryotes; the median value is 17 fmol SO₄²⁻ cell⁻¹ d⁻¹ (range: 0.9-434). Reference 38 reports an average cell-specific methane production rate (apart from super-optimal temperatures) for *Methanocaldococcus str. JH146* of 0.39 pmol cell⁻¹ h⁻¹ (9400 fmol CH₄ cell⁻¹ d⁻¹).

A more accurate representation of these references would be: “Methanogenic and sulfate reducing microorganisms in colder sediment (<15°C) operate at cell-specific metabolic rates of <1 fmol (CH₄ or SO₄²⁻) cell⁻¹ d⁻¹, which may decrease steeply³¹ or remain constant³³ with depth. ... By contrast, the median respiration rate for 32 pure cultures of sulfate-reducing microorganisms is 17 fmol SO₄²⁻ cell⁻¹ d⁻¹ (range: 0.9-434)³⁶ and can reach several hundred fmol SO₄²⁻ cell⁻¹ d⁻¹ for (hyper-)thermophiles such as *Archaeoglobus fulgidus*.³⁷ Respiration rates for (hyper-)thermophilic methanogens such as *Methanocaldococcus str. JH146* can reach nearly ten thousand fmol CH₄ cell⁻¹ d⁻¹.³⁸”

I realize that shifting the rule-of-thumb by an order-of-magnitude (i.e., cold sediments <1 fmol cell⁻¹ d⁻¹, pure cultures typically >>10 fmol cell⁻¹ d⁻¹) does not affect the claim that potential cell-specific sulfate reduction and methane production rates in deep, hot sediments (7.5 fmol cell⁻¹ d⁻¹) are similar to rates in laboratory cultures. However, while reading reference 36 I realized the obvious: temperature must be considered when comparing rates of metabolism. The potential cell-specific rate measured for sulfate-reducers from sub-seafloor sediment incubated at ~80°C with saturating concentrations of acetate is similar to *Desulfofrigus oceanense* (7.6 fmol cell⁻¹ d⁻¹; reference 36) isolated from Arctic sediment and incubated at 9°C with 20 mM acetate. In contrast, *Archaeoglobus fulgidus* isolated from a submarine hot spring and incubated at 80°C with 20 mM lactate has a cell-specific sulfate-reduction rate of 63.8 fmol cell⁻¹ d⁻¹ (reference 36; reference 37 reports a cell-specific sulfate-reduction rate of 336 fmol cell⁻¹ d⁻¹ for the same species grown at 82-84°C). Thus, sulfate reducers in the deep, hot subseafloor biosphere subsist with unexpectedly *low* cell-specific rates of catabolism compared to pure cultures at the same temperature. This provides a simple explanation for the low cell density in these sediments: The power available for growth is equal to the power from the catabolic reaction (rate × ΔG) less the power used for functions other than growth (including maintenance). Since maintenance increases with temperature, cell growth rates (and presumably cell-densities) will decrease with increasing temperature if the rate of catabolism fails to keep up with maintenance. It is important to note that the possibility that metabolism in the deep, hot subsurface biosphere is unexpectedly slow does not diminish the significance of this work.

Author's Response:

We thank the reviewer for pointing out that we oversimplified our comparison with laboratory cultures by not taking temperature into account. We revised our discussion to emphasize that there is a difference between metabolic rates of thermophilic and non-thermophilic laboratory cultures. Further, we followed the reviewer's suggestions to improve the representation of literature values of cell-specific rates, as described in the following.

Based on the reviewer's suggestion we now cite rates in sediments at colder temperatures, as follows:

"Methanogenic and sulfate reducing microorganisms in cold sediment (<15°C) generally operate at cell-specific metabolic rates of <1 fmol (SO₄²⁻ or CH₄) cell⁻¹ d⁻¹ in the upper sediment"

With respect to the Beulig et al. 2019 reference, the reviewer correctly pointed out that the cell-specific rates remained constant over depth in this study from Arhus Bay (Baltic Sea). However, it is important to note that these sediments were organic-rich and sampled at shallow sediment depths (<1.5 m). For deeper sediment, however, data reported in the cited literature shows that cell-specific rates are indeed much lower (>2,000-fold) as highlighted in the revised text:

"The cell-specific rates of sulfate reducing microorganisms may decrease to <0.01 fmol SO₄²⁻ cell⁻¹ d⁻¹ below 1 mbsf, and to <0.001 fmol SO₄²⁻ cell⁻¹ d⁻¹ in the deep sub-seafloor"

In our comparison to laboratory cultures, we now make a distinction between pure cultures at psychrophilic to mesophilic temperatures (<45°C) and (hyper-)thermophilic cultures, as follows:

"By contrast, respiration rates for pure cultures of sulfate-reducing or methanogenic microorganisms are typically >1 fmol (SO₄²⁻ or CH₄) cell⁻¹ d⁻¹ at psychrophilic to mesophilic temperatures (<45°C), and can reach several hundred to thousand fmol (SO₄²⁻ or CH₄) cell⁻¹ d⁻¹ for (hyper-)thermophiles"

It is further worth noting that potential cell-specific rates of uncultured sulfate-reducing microorganisms in advective, hydrothermal surface sediments (0.3-0.5 fmol SO₄²⁻ cell⁻¹ d⁻¹ at >60°C; Rusch & Amend, 2008. *Microbial ecology*, 55(4), pp.723-736) are in the same range as found in our study. However, laboratory cultures, enriched from the same sediment, showed two orders of magnitude higher cell-specific rates (46-82 fmol SO₄²⁻ cell⁻¹ d⁻¹), likely due to excess supply of substrates in culture treatments.

In summary, we conclude that potential cell-specific rates reported in the present study approach the range of active surface sediments and (non-thermophilic) laboratory cultures,

and that they are unexpectedly high compared to the cited cold deep sub-seafloor with a >2,000-fold difference.

Reviewer's Comment:

In conclusion, I hoped to make three points. First, the calculations in my review did not agree with the calculations and assumptions in the manuscript. Second, the characterization of literature values of cell-specific rates is not accurate. (While the errors do not affect the conclusions of this paper, errors in the literature tend to be repeated, and with enough repetition become accepted as fact to the detriment of the scientific community.) Third, temperature must be considered when comparing rates of metabolism.

Author's Response:

We like to thank the reviewer again for this in-depth review and discussion, which pushed us to double check and refine our calculations and to clarify our statements. We are confident that our new calculation of sulfate consumption is now in line with the reviewer's approach and that numbers from cited literature are presented correctly. We further hope that we were able to convince the reviewer that the here reported cell-specific rates are indeed unexpectedly high compared to the deep cold biosphere.

Manuscript: NCOMMS-21-12665

Title: Rapid Metabolism Fosters Microbial Survival in the Deep, Hot, Subseafloor Biosphere

Authors: Beulig et al.

The authors have addressed all my comments. I have two final suggestions that I offer in the spirit that they could possibly boost the impact of the paper. In other words, I don't want to see this paper again (that is, until it comes out in Nature Communications).

1. Lines 38-40: “The small microbial community subsisted with *unexpectedly high* potential cell-specific rates of energy metabolism, which *approach* the range of active surface sediments and laboratory cultures” (italics added).

The terms in italics are subjective and not clearly qualified. For example, it is correct to state that the cell-specific sulfate reduction rates in hot (80°C) sediments with saturating substrate concentrations ($>0.2 \text{ fmol}\cdot\text{cell}^{-1}\cdot\text{d}^{-1}$) are *high* compared to rates in cold ($<15^\circ\text{C}$) sediments with limiting substrate concentrations ($>0.01 \text{ fmol}\cdot\text{cell}^{-1}\cdot\text{d}^{-1}$). But it is also correct to state that the cell-specific sulfate reduction rates in hot (80°C) sediments with saturating substrate concentrations are *low* compared to thermophilic sulfate reducers ($\sim 80^\circ\text{C}$) cultured with saturating substrate concentrations ($336 \text{ fmol}\cdot\text{cell}^{-1}\cdot\text{d}^{-1}$). The observation that cell-specific sulfate reduction rates at 80°C (with mM acetate) are higher than cell-specific rates at $<15^\circ\text{C}$ (with μM acetate) is *not unexpected* (at least to me; the Arrhenius equation predicts that cell-specific rates increase 100-fold between 15-80°C for $E_a = 60 \text{ kJ/mol}$; the Monod equation predicts that rates increase 8-fold between 10 μM and 10 mM acetate for $K_S = 70 \mu\text{M}$). However, it is *unexpected* (at least to me) that cell-specific sulfate reduction rates in hot (80°C) sediments with saturating substrate are three orders-of-magnitude slower than thermophilic sulfate reducers under comparable substrate concentrations.

Another example of subjective language is the term *approach*: whether >0.2 *approaches* <1 (active surface sediments) and >1 (psychrophilic and mesophilic laboratory cultures) is a matter of opinion.

To improve clarity, I suggest that lines 38-40 be rephrased and the subjective terms replaced with concrete values. I provide an example below.

“The small microbial community subsisted with potential cell-specific rates of energy metabolism of $>0.2 \text{ fmol}\cdot\text{cell}^{-1}\cdot\text{d}^{-1}$. For comparison, microbial communities that subsist in cold, acetate-deplete sediment generally operate at $<1 \text{ fmol}\cdot\text{cell}^{-1}\cdot\text{d}^{-1}$ in the upper sediment, decreasing by orders-of-magnitude in the cold, deep sub-seafloor. For pure cultures, cell-specific rates range from $>1 \text{ fmol}\cdot\text{cell}^{-1}\cdot\text{d}^{-1}$ for psychro- to mesophilic organisms and up to $10^2\text{-}10^3 \text{ fmol}\cdot\text{cell}^{-1}\cdot\text{d}^{-1}$ for (hyper-)thermophiles.”

The reason I am pushing the authors to consider avoiding phrases like “rapid” and “unexpectedly fast” is that it may have led them to downplay a most interesting aspect of this study: rates of metabolism in the deep, hot subseafloor biosphere are orders of-magnitude slower than rates of metabolism in related hyperthermophiles cultured under similar conditions.

2. Lines 137-149: I concur that for sediments with a formation factor of 14 ($\theta^2 = 4.9$), diffusion over a time period of 400,000 y would have minimal effect on the sulfate profile below 750 mbsf. Hence, it is justified to treat the lower 450 m of the sediment column as a closed system. In my first review, I wrote that for a closed system, the average rate needed to reduce sulfate from 28 mM to ~6 mM (the average concentration below 750 m) is simply:

$$\frac{22 \mu\text{mol}}{\text{cm}_{\text{iw}}^3} \cdot \frac{0.35 \text{ cm}_{\text{pw}}^3}{\text{cm}_{\text{ws}}^3} \cdot \frac{10^6 \text{ pmol}}{\mu\text{mol}} \cdot \frac{1}{400,000 \text{ y}} \cdot \frac{\text{y}}{365 \text{ d}} = 0.05 \text{ pmol} \cdot \text{cm}_{\text{ws}}^{-3} \cdot \text{d}^{-1}.$$

I suggested that this is a more direct approach to assess the plausibility of the measured potential rates (~0.3 vs. 0.05 pmol · cm_{ws}⁻³ · d⁻¹) than comparing the time it would take to completely deplete sulfate to the time that has elapsed since the rapid burial event (90,000 y vs. 400,000 y). Both approaches have the same result, but in my view directly comparing rates will make it easier for readers to assess.

My final suggestion: rather than state that “there is *remarkably good agreement* between the two methods” (italics added), let readers decide for themselves whether the agreement is remarkable. Here is an alternative phrasing that avoids subjective terms:

“The difference between the spatially-averaged *in-situ* sulfate depletion rate of 0.05 pmol · cm_{ws}⁻³ · d⁻¹ and the measured potential sulfate reduction rate of ~0.3 pmol · cm_{ws}⁻³ · d⁻¹ could be attributed to the time it takes for rates to increase following burial heating, and may also be affected by spatial variation in cell abundances.”

Response to Reviewer #3

We thank Reviewer #3 again for the constructive and helpful comments to our revised manuscript.

Reviewer's Comment:

The authors have addressed all my comments. I have two final suggestions that I offer in the spirit that they could possibly boost the impact of the paper. In other words, I don't want to see this paper again (that is, until it comes out in Nature Communications).

1. Lines 38-40: "The small microbial community subsisted with *unexpectedly high* potential cell-specific rates of energy metabolism, which *approach* the range of active surface sediments and laboratory cultures" (italics added).

The terms in italics are subjective and not clearly qualified. For example, it is correct to state that the cell-specific sulfate reduction rates in hot (80°C) sediments with saturating substrate concentrations ($>0.2 \text{ fmol}\cdot\text{cell}^{-1}\cdot\text{d}^{-1}$) are *high* compared to rates in cold ($<15^\circ\text{C}$) sediments with limiting substrate concentrations ($>0.01 \text{ fmol}\cdot\text{cell}^{-1}\cdot\text{d}^{-1}$). But it is also correct to state that the cell-specific sulfate reduction rates in hot (80°C) sediments with saturating substrate concentrations are *low* compared to thermophilic sulfate reducers ($\sim 80^\circ\text{C}$) cultured with saturating substrate concentrations ($336 \text{ fmol}\cdot\text{cell}^{-1}\cdot\text{d}^{-1}$). The observation that cell-specific sulfate reduction rates at 80°C (with mM acetate) are higher than cell-specific rates at $<15^\circ\text{C}$ (with μM acetate) is *not unexpected* (at least to me; the Arrhenius equation predicts that cell-specific rates increase 100-fold between 15-80°C for $E_a = 60 \text{ kJ/mol}$; the Monod equation predicts that rates increase 8-fold between 10 μM and 10 mM acetate for $K_S = 70 \mu\text{M}$). However, it is *unexpected* (at least to me) that cell-specific sulfate reduction rates in hot (80°C) sediments with saturating substrate are three orders-of-magnitude slower than thermophilic sulfate reducers under comparable substrate concentrations.

Another example of subjective language is the term *approach*: whether >0.2 *approaches* <1 (active surface sediments) and >1 (psychrophilic and mesophilic laboratory cultures) is a matter of opinion.

Author's Response:

We agree with the reviewer and removed suggestive language throughout the manuscript.

Reviewer's Comment:

To improve clarity, I suggest that lines 38-40 be rephrased and the subjective terms replaced with concrete values. I provide an example below.

"The small microbial community subsisted with potential cell-specific rates of energy metabolism of $>0.2 \text{ fmol}\cdot\text{cell}^{-1}\cdot\text{d}^{-1}$. For comparison, microbial communities that subsist in cold, acetate-deplete sediment generally operate at $<1 \text{ fmol}\cdot\text{cell}^{-1}\cdot\text{d}^{-1}$ in the upper sediment, decreasing by orders-of-magnitude in the cold, deep sub-seafloor. For pure cultures, cell-specific rates range from $>1 \text{ fmol}\cdot\text{cell}^{-1}\cdot\text{d}^{-1}$ for psychro- to mesophilic organisms and up to 10^2 - $10^3 \text{ fmol}\cdot\text{cell}^{-1}\cdot\text{d}^{-1}$ for (hyper-)thermophiles."

Author's Response:

The lines 38-40 refer to the abstract. Adding the alternative text suggested by the reviewer would extend the abstract length substantially above the word limit. We further believe that detailed numbers should not be presented in a Nature Communication abstract, which is supposed to aim at a broader audience. Providing these details in the main text seems sufficient. We refined the abstract to remove unspecific language where appropriate.

Reviewer's Comment:

2. Lines 137-149: I concur that for sediments with a formation factor of 14 ($\theta^2 = 4.9$), diffusion over a time period of 400,000 y would have minimal effect on the sulfate profile below 750 mbsf. Hence, it is justified to treat the lower 450 m of the sediment column as a closed system. In my first review, I wrote that for a closed system, the average rate needed to reduce sulfate from 28 mM to ~6 mM (the average concentration below 750 m) is simply:

$$\frac{22 \mu\text{mol}}{\text{cm}_{\text{iw}}^3} \cdot \frac{0.35 \text{ cm}_{\text{pw}}^3}{\text{cm}_{\text{ws}}^3} \cdot \frac{10^6 \text{ pmol}}{\mu\text{mol}} \cdot \frac{1}{400,000 \text{ y}} \cdot \frac{\text{y}}{365 \text{ d}} = 0.05 \text{ pmol} \cdot \text{cm}_{\text{ws}}^{-3} \cdot \text{d}^{-1}.$$

I suggested that this is a more direct approach to assess the plausibility of the measured potential rates (~0.3 vs. 0.05 $\text{pmol} \cdot \text{cm}_{\text{ws}}^{-3} \cdot \text{d}^{-1}$) than comparing the time it would take to completely deplete sulfate to the time that has elapsed since the rapid burial event (90,000 y vs. 400,000 y). Both approaches have the same result, but in my view directly comparing rates will make it easier for readers to assess.

Author's Response:

We very much appreciate the calculation and integrated it into the Methods section (see under *Calculation of pore water sulfate utilization and sediment organic carbon oxidation*) to provide an alternative check for the plausibility of the determined rates.

Reviewer's Comment:

My final suggestion: rather than state that “there is *remarkably good agreement* between the two methods” (italics added), let readers decide for themselves whether the agreement is remarkable. Here is an alternative phrasing that avoids subjective terms:

“The difference between the spatially-averaged *in-situ* sulfate depletion rate of 0.05 $\text{pmol} \cdot \text{cm}_{\text{ws}}^{-3} \cdot \text{d}^{-1}$ and the measured potential sulfate reduction rate of ~0.3 $\text{pmol} \cdot \text{cm}_{\text{ws}}^{-3} \cdot \text{d}^{-1}$ could be attributed to the time it takes for rates to increase following burial heating, and may also be affected by spatial variation in cell abundances.”

Author's Response:

Thank you for this suggestion. We removed "remarkably" to avoid suggestive language.